# Subobject-level Image Tokenization

**Delong Chen** [1 2]  **Samuel Cahyawijaya** [2]  **Jianfeng Liu** [3]  **Baoyuan Wang** [4]  **Pascale Fung** [1 2]

## Abstract

Patch-based image tokenization ignores the morphology of the visual world, limiting effective and efficient learning of image understanding. Inspired by subword tokenization, we introduce *subobject*-level adaptive token segmentation and explore several approaches, including superpixel, SAM, and a proposed **E**fficient and **P**an**O**pti**C** (EPOC) image tokenizer. Our EPOC combines boundary detection–a simple task that can be handled well by a compact model–with watershed segmentation, which inherently guarantees no pixels are left unsegmented. Intrinsic evaluations across 5 datasets demonstrate that EPOC's segmentation aligns well with human annotations of both object- and part-level visual morphology, producing more monosemantic tokens and offering substantial efficiency advantages. For extrinsic evaluation, we designed a token embedding that handles arbitrary-shaped tokens, and trained VLMs with different tokenizers on 4 datasets of object recognition and detailed captioning. The results reveal that subobject tokenization enables faster convergence and better generalization while using fewer visual tokens. Project website: https://github.com/ChenDelong1999/subobjects.

## 1. Introduction

Partitioning the raw observational data stream into a manageable number of segments is often beneficial for learning, particularly when nearby elements in the data stream exhibit strong correlations and therefore high redundancy (Cover, 1999; Barlow et al., 1961; Biederman, 1987; Marr, 2010; Delétang et al., 2024). In Computer Vision (CV), nearby pixels are often highly correlated (He et al., 2022; Simon-

[1]Meta FAIR Paris [2]The Hong Kong University of Science and Technology [3]Alibaba Group [4]Zillow. Correspondence to: Pascale Fung <pascale@ece.ust.hk>, Delong Chen <delong.chen@connect.ust.hk>.

*Proceedings of the 42$^{nd}$ International Conference on Machine Learning*, Vancouver, Canada. PMLR 267, 2025. Copyright 2025 by the author(s).

celli & Olshausen, 2001). Direct modeling on pixels (Chen et al., 2020; Nguyen et al., 2024) tends to be computationally inefficient, especially for high-resolution images. The widely adopted approach is patch-based tokenization (Dosovitskiy et al., 2021), which divides images into fixed-size square patches, each treated as a single token. Although being straightforward and efficient, the *non-adaptive* nature of patch-based tokenization prohibits its ability to capture the underlying morphological structure of the image (Palmer, 1977). Consequently, larger patches fuse multiple semantics within a single token (*polysemanticity*), an undesirable property that hinders effective learning of token representations. On the other hand, smaller patches exhibit greater *monosemanticity* but lead to an excessive number of tokens, resulting in computational inefficiency.

In Natural Language Processing (NLP), similar challenges of input redundancy also exist, and the solution is to merge characters that frequently co-occur into individual tokens (Sennrich et al., 2016; Kudo, 2018; Gastaldi et al., 2024). Sennrich et al. (2016) demonstrates that a statical-based grouping approach can produce subword tokens that identifies the morphological structure, *e.g.,* the word *sweetish* in English can be chunked into "*sweet*" and "*-ish*", while the word "*süßlich*" in German, can be chunked into "*süß*" and "*-lich*". This yields better monosemanticity of tokens compared to other alternatives such as word-based or character-based tokenization methods which improves generalization to rare or out-of-vocabulary words by enabling semantic composition of subwords (Wang et al., 2018; Wilie et al., 2020; Mielke et al., 2021), while also guaranteeing efficiency by significantly reducing sequence length (Song et al., 2021; Muller et al., 2022; Cahyawijaya et al., 2024).

Patch tokenization on images is analogous to "character chunk tokenization" on text, which is conterintuitive and rarely adopted in practice. We hypothesize that *adaptive* token segmentation, as opposed to static patch tokenization, can facilitate better learning of image understanding. To test this, we explore both **object-level** tokenization, which is akin to word-level tokenization, and **subobject-level** tokenization, resembling subwords in images. For subobject segmentation, a promising method is SAM's *"segment everything"* mode (Kirillov et al., 2023), but it suffers from computational inefficiency and the presence of unsegmented regions. We propose **E**fficient and **P**an**O**pti**C** (EPOC) to

addresses these two key limitations. EPOC integrates boundary detection and watershed segmentation (Vincent & Soille, 1991). The boundary detection formulation (instead of mask generation) simplifies the learning task (Canny, 1986) and enables us to squeeze the model size to **3.7M parameters** from SAM's 641M while achieving a similar level of segmentation quality[1]. Moreover, the revisited watershed algorithm ensures comprehensive segmentation while also providing flexible control over segmentation granularity through a single scalar hyperparameter. This design improves inference efficiency by making the cost independent of the number of masks (*i.e.,* $O(1)$ complexity).

Studies on text tokenization in NLP typically involve intrinsic evaluation measuring tokenizer's innate qualities, and extrinsic evaluation measuring its impact on downstream task performance (Gowda & May, 2020; Zouhar et al., 2023; Goldman et al., 2024). Following this practice, we also measure both aspects for a holistic evaluation. **Intrinsic evaluation** encompasses three key aspects: *1) morphology*– whether the segmentation align with semantic boundaries, *2) monosemanticity*–whether individual token avoids covering multiple semantics, and *3) efficiency*–how much additional computational overhead are introduced. Results on five datasets covering both object- and subobject-level annotations reveals that both SAM and EPOC yields strong morphology alignment and token monosemanticity, while EPOC enjoys significant advantage on efficiency.

For **extrinsic evaluation**, we first design a vision-language model (VLM) (Liu et al., 2023) architecture to incorporate adaptive token segmentation which produces dynamic spatial arrangement and non-regular shapes, and train a series of models with different image tokenizers. Results on ImageNet-1k (Deng et al., 2009), ShareGPT4V (Chen et al., 2024), Pixmo-cap (Deitke et al., 2024), as well as a new dataset CLEVR-cap generated from CLEVR (Johnson et al., 2017) indicates that VLMs using subobject-level image tokenization enjoy faster convergence and better generalization with much fewer number of visual tokens. Such advantage can be observed across different LLMs and visual embeddings. We also show that EPOC-based method holds more robustness to dropping long-tail small tokens, which allow us to further reduce sequence length. To summarize, we make three core contributions in this paper:

- We introduce EPOC, a novel subobject tokenization that integrates boundary detection and watershed segmentation to achieve efficient panoptic segmentation.
- We compare different adaptive token segmentation methods via extensive intrinsic evaluation. EPOC provides comparable segmentation quality while being

---

[1]The boundary detection model was referred to as the Direct Segment Anything Model (DirectSAM) in our initial arXiv versions. Here EPOC = DirectSAM + Watershed

much more efficient.

- We show that subobject-level image tokenizers facilitate faster VLMs convergence, improves their generalization, and achieves better token efficiency, outperforming patch and object toknizers.

## 2. Preliminaries

### 2.1. Problem Formulation

Let $\mathbf{X} \in \mathbb{R}^{H \times W \times 3}$ represent an input image and $\mathbf{Y}$ represent the target label, the task of image understanding is to learn a function $f$ that maps $\mathbf{X}$ to $\mathbf{Y}$. When using sequence models such as Transformers, this mapping is achieved by processing and generating sequences of tokens derived from $\mathbf{X}$ and $\mathbf{Y}$, *i.e.,* $f([\mathbf{x}_1, \mathbf{x}_2, \ldots, \mathbf{x}_N]) = [\mathbf{y}_1, \mathbf{y}_2, \ldots, \mathbf{y}_m]$, where each vector $\mathbf{x}_i \in \mathbb{R}^d$ is an individual visual token and $\{\mathbf{y}_i\}_{i=1}^m$ are text tokens. The task of *image tokenization* is to transform the high-dimensional raw input $\mathbf{X}$ into a compact set of visual tokens, *i.e.,* $\mathbf{X} \mapsto [\mathbf{x}_1, \mathbf{x}_2, \ldots, \mathbf{x}_N]$.

The process of image tokenization consists of token segmentation and token embedding. Token segmentation aims to produce a token index map $\mathbf{M} \in \{0, \ldots, N-1\}^{H \times W}$ from $\mathbf{X}$, where each element in this map assigns the corresponding pixel to one of $N$ tokens. All pixels assigned to the same token index are grouped together as individual visual tokens. We are interested in developing good token segmentation that enables $f$ to better approximate $\mathbf{X} \mapsto \mathbf{Y}$ (*i.e.,* faster convergence and better generalization). Furthermore, since the computational cost of $f$ usually heavily depends on the number of input tokens, the mapping $\mathbf{M}$ that has smaller $N$ is preferred if the performance of $f$ is the same. In this paper, we propose and implement different token segmentation approaches for generating $\mathbf{M}$, and learn $f : \mathbf{X} \mapsto \mathbf{Y}$ with each of them under controlled settings.

### 2.2. Patch-based Image Tokenization

Patch tokenization divides the image into $N = \mathrm{p} \times \mathrm{p}$ non-overlapping square patches of fixed size, where $\mathrm{p}$ controls the number of patches per side. Its effectiveness lies in leveraging the spatial inductive bias that neighboring pixels exhibit high correlations. However, the underlying assumption of patch tokenization can be overly strong since visual semantics are not usually distributed in a grid-like structure. The violation of such unrealistic assumption leads to the issues of either **token polysemanticity** or **token redundancy**. Large patches in crowded regions often encompass multiple semantics. While a model $f$ can learn to represent such polysemantic tokens, the *unique mixture of semantics is unlikely to appear frequently* in the training data, leading to limited sample exposure and insufficient learning. This issue is analogous to the challenge of rare words in word-based text tokenization in NLP, which can be addressed by

increasing the segmentation granularity. The solution for vision models that shares the similar spirit is to lower the patch size, as adopted by many recent large ViTs. However, this introduces redundancy. For instance, a large region of clean sky can be divided into excessive number of tokens, wasting computation resources.

## 3. Adaptive Token Segmentation

As patch-based tokenization struggles with the conflict between token polysemanticity and token redundancy, this section introduces adaptive token segmentation methods, which include both segmenting images into object instances (§3.1) and subobject entities (§3.2).

### 3.1. Object-level Image Tokenization

A straightforward approach that enables adaptive tokenization is to group pixels by object instances. Here, "objects" refers broadly to both discrete "thing" instances (*e.g.,* cats, humans, cars) and amorphous "stuff" regions (*e.g.,* sky, grass, water). This approach is promising due to the natural coherence of objects in the real world, where their constituent pixels exhibit high internal correlations as they tend to move as a whole. Object-level image tokenization can be implemented using panoptic segmentation (Kirillov et al., 2019) models, where the generated "label map" can be directly utilized as token segmentation $\mathbf{M}$.

Panoptic segmentation models are typically trained on the COCO (Lin et al., 2014) and ADE20K (Zhou et al., 2019) datasets, which respectively include 133 and 150 annotated classes of common objects and regions. However, the number of object types in the real world far exceeds this scale (Wang et al., 2023). A significant limitation of these models is their reliance on the **fixed vocabulary of object categories**, which poses challenges for generalization to out-of-vocabulary objects. In addition, image understanding tasks are not always object-centric (Tong et al., 2024). When the target $\mathbf{Y}$ involves information of object parts, object-level tokenization will still suffer from polysemanticity.

### 3.2. Subobject-level Image Tokenization

In NLP, subword tokenization has demonstrated superior performance in language modeling compared to word-based tokenization. Inspired by subwords and with an analogy of object in image are akin to words in sentences, we introduce the notion of *subobject*-level image tokenization. Subobjects represent an intermediate *level* situated between objects and pixels, similar to the concept of subwords which is also an umbrella covers various methods.

Subobject entities are semantically coherent and functionally meaningful units, encompassing object parts, sub-parts, and finer subdivisions. The potential of subobject tokenization not only come from its similarity with subword tokenization, but also from its alignment with the **Recognition-by-Components** (Biederman, 1987) theory in cognitive science, which posits that human recognize objects through their constituent parts (Tversky & Hemenway, 1984; Dehaene et al., 2022). Moreover, in previous CV studies, part-based image recognition (Felzenszwalb & Huttenlocher, 2005) has also demonstrated improved robustness (Li et al., 2023; Sitawarin et al., 2023; Li et al., 2024) and sample efficiency (Lake et al., 2015). In the following, we introduce three subobject segmentation approaches that we explore.

**3.2.1. Superpixel Segmentation.** Superpixel segmentation is a classical type of method that groups adjacent pixels into local clusters. These clusters, termed superpixels, are formed based on pixel properties such as color intensity. This typically results in irregularly shaped segments that are smaller and more detailed than entire objects, and as such, superpixels are naturally categorized within the subobject level. Over decades, many superpixel segmentation methods have been developed. We consider the $k$-means-based Simple Linear Iterative Clustering (SLIC) (Achanta et al., 2012) method. The $\mathtt{k}$ for $k$-means is the key hyperparameters that allow control over the granularity of segmentation.

**3.2.2. Segment Anything Models.** Segment Anything Models (SAM) are trained on the large SA-1B dataset, which contains 11 million images and 1 billion mask annotations of varying granularities, encompassing objects, parts, and subparts (Kirillov et al., 2023). The SAM architecture combines an image encoder and a prompt-conditioned mask decoder, enabling *segment anything* according to the given prompt. To generate a comprehensive segmentation for a whole image, a regular grid of point prompts is applied to the decoder, producing masks corresponding to each point prompt. These masks are subsequently filtered and merged to yield the final *segment everything* results.

While SAM demonstrates strong performance across diverse image segmentation benchmarks, its effective application in image tokenization is hindered by the following two significant limitations: **Efficiency**: To perform *segment everything* on a single image with a default grid of $32 \times 32$ point prompts requires 1,024 times of forward passes through the mask decoder. Although the mask decoder has been made lightweight, and improvements such as reducing the encoder size (Zhao et al., 2023) and reducing the number of prompts (Zhang et al., 2023) have been introduced, the complexity remains $O(N)$, where $N$ is the number of masks to be generated. **Comprehensiveness**: As shown in Fig. 1, the result of SAM's *segment everything* mode is not guaranteed to be panoptic. SAM's design of generating each mask independently introduces potential gaps between individual segments, or certain background regions may remain unsegmented. While these gaps or unsegmented areas can be

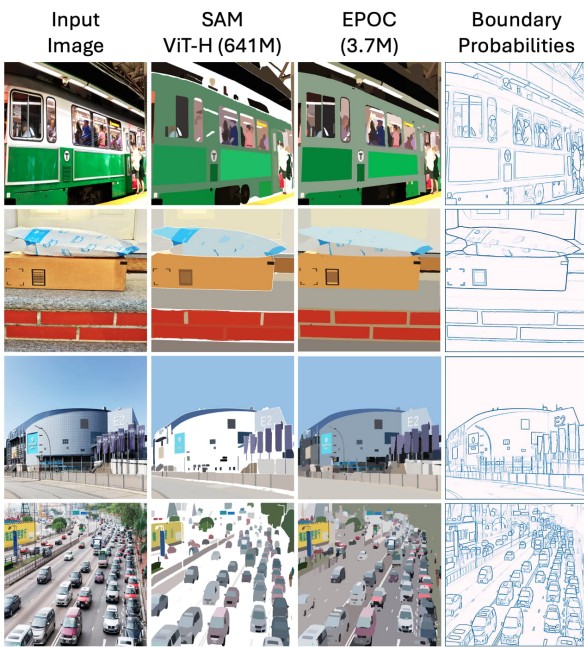

Input Image  SAM ViT-H (641M)  EPOC (3.7M)  Boundary Probabilities

*Figure 1.* **Comparing SAM and our EPOC on SA-1B images**. The design of independent mask decoding makes SAM often leave thin gaps between segments or background regions unsegmented. Our EPOC inherently guarantees complete coverage while also improves computational efficiency.

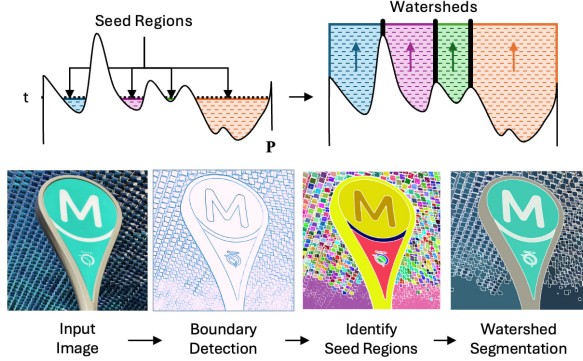

*Figure 2.* **Proposed EPOC**. A boundary probability map $\mathbf{P} \in [0, 1]^{H \times W}$ is predicted from the input image $\mathbf{X} \in \mathbb{R}^{H \times W \times 3}$ and treated as a topographical surface. The watershed segmentation begins by identifying basins in $\mathbf{P}$ as seed regions (labeled in different colors) with a threshold $\mathtt{t}$. A "flooding" process then progresses until the entire $\mathbf{P}$ is submerged. When seed regions meet during flooding, "watersheds" are formed to separate them.

grouped into a single token to avoid information loss, the resulting tokens often suffer from high *polysemanticity*.

**3.2.3. Proposed EPOC.** We propose Efficient and Comprehensive Image Tokenization (EPOC), a novel *segment everything* approach that ensures comprehensive segmentation while significantly improving computational efficiency. Unlike SAM's prompt-conditioned mask decoding, EPOC adopts a **boundary detection** approach, predicting a pixel-wise boundary probability map $\mathbf{P} \in [0, 1]^{H \times W}$ that outlines subobject entities across the entire image. This approach avoids the overhead of generating individual masks and benefits from the simplicity of boundary detection. Tasks of similar difficulty have historically been effectively handled by non-parametric algorithms (Canny, 1986) and early layers of CNNs (Zeiler & Fergus, 2014). The task simplicity allow us to employ an extremely compact model. We empirically found a lightweight SegFormer-b0 (Xie et al., 2021) with only 3.7M parameters learns well from SA-1B.

Based on the predicted boundary probability map $\mathbf{P}$, a naïve approach to derive the token segmentation map $\mathbf{M}$ might involve thresholding the map and applying connected component analysis (Wu et al., 2009). However, such methods leave boundary pixels unsegmented, leading to potential information loss or polysemanticity when these pixels are merged. Instead, we revisit the **watershed algorithm** (Vin-

cent & Soille, 1991), which naturally ensures comprehensiveness. As shown in Fig. 2, the method treats the predicted $\mathbf{P}$ metaphorically as a topographical surface, where higher probabilities correspond to elevated regions like peaks or ridges and lower probabilities represent basins. The algorithm simulates a gradual *"flooding"* of these basins, with boundary of segmentation masks are generated acting as barriers that prevent water from merging between different basins. Watershed typically requires **seed regions** to begin the "flooding" process, which we obtain by applying a threshold $\mathtt{t}$ to the boundary map. Pixels below $\mathtt{t}$ are assumed to be within object interiors and serve as seeds. By tuning $\mathtt{t}$, one can seamlessly control the granularity of the resulting segmentation—higher $\mathtt{t}$ merges seeds into larger segments, while lower $\mathtt{t}$ retains finer details. Fig. 10 in Appendix provide an example illustrating such flexibility.

EPOC achieves its computational efficiency through three advantages. **First**, the boundary probability predictor operates on the entire image in a single forward pass, resulting in a complexity of $O(1)$ which is independent to the number of tokens $N$. **Second**, the compactness of SegFormer-b0 further reduces GPU memory usage, allowing for large batch sizes or multi-process parallelism. **Third**, the subsequent watershed algorithm is non-parametric and runs efficiently on CPUs (Kornilov & Safonov, 2018). These features together make EPOC an efficient image tokenizer.

# 4. Intrinsic Evaluations

## 4.1. Evaluation Setup

This section evaluates innate properties of token segmentation models (which take image $\mathbf{X}$ to generate the index

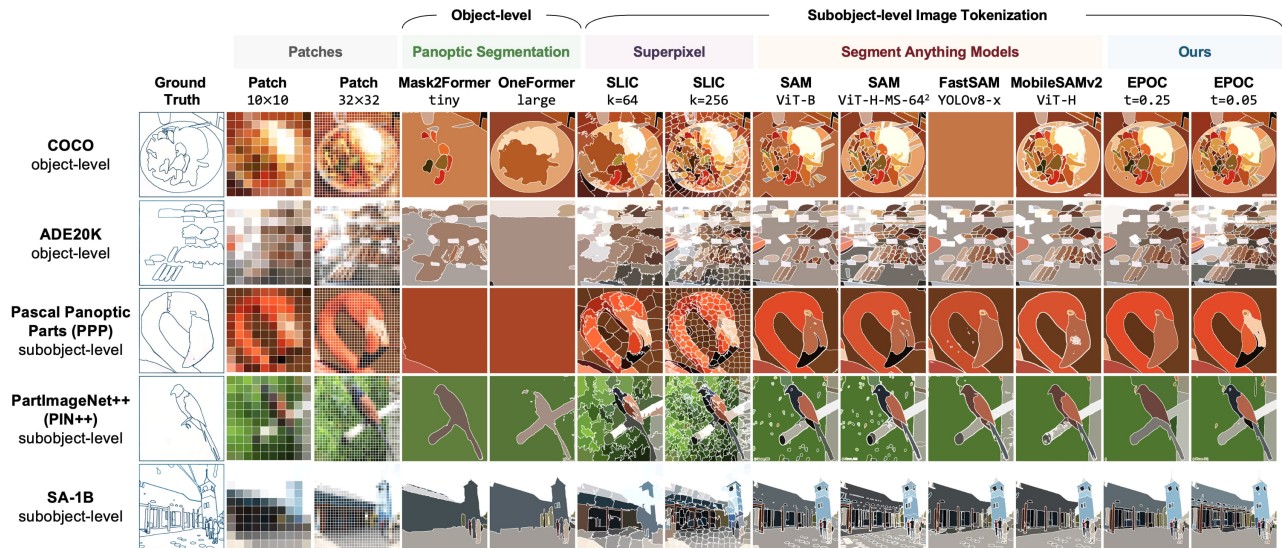

*Figure 3.* **Intrinsic evaluation dataset examples and token segmentation results.** **Object-level** tokenization based on panoptic segmentation suffers from out-of-vocabulary problem. **Superpixel** segmentation relies on bottom-up pixel grouping, which limits its ability to capture underlying structures. The **SAM** model and its variants generally provide reasonable token segmentation. The quality and style of the segmentation generated by our **EPOC** closely match those of SAM, while utilizing a significantly smaller model size.

map $\mathbf{M}$ (§3) without training any downstream models (*e.g.,* VLMs). **Models.** We compare three categories of candidates: **patch-based**: square patches with patch size p varying from 2 to 32; **object-level**: Mask2Former (Cheng et al., 2022) and OneFormer (Jain et al., 2023) trained on COCO and ADE20K. **subobject-level**: SLIC superpixel (Achanta et al., 2012), SAM (Kirillov et al., 2023), FastSAM (Zhao et al., 2023), MobileSAMv2 (Zhang et al., 2023), and the proposed EPOC method. **Datasets.** We conduct evaluations on five datasets, encompassing different annotation granularities, with COCO's COCONut relabeled validation split (Deng et al., 2024) and ADE-20K (Zhou et al., 2019) validation split provide **object-level** annotations, and Pascal Panoptic Parts (PPP) (de Geus et al., 2021), PartImageNet++ (PIN++) (Li et al., 2024) and SA-1B (Kirillov et al., 2023) consist **subobject-level** annotations. Fig. 3 provide visualization of converted ground truth boundary (leftmost) and token segmentation generated by different tokenizers. We provide richer details in the Appendix A and B.

### 4.2. Results and Discussions

**Alignment to Morphology.** We first measure how well different token segmentation methods capture the semantic structures of images. Although the human annotations are provided as mask, traditional mask-based metrics are unsuitable for this evaluation due to their incompatibility with class-agnostic and mixed-granularity segmentation (as we have both object- and subobject-level models and annotations). Instead, we adopt boundary precision-recall metrics, which are widely used in boundary / edge / contour

detection studies (Arbeláez et al., 2011; Yamagiwa et al., 2024). A token segmentation that accurately captures the semantic structure of an image should align well with the ground truth boundaries, achieving both high precision and recall. The top row of Fig.4 presents the results for an input resolution of 768px. The evaluation confirms that **patch-based** tokenization exhibits very low alignment with image morphological structures, whereas all adaptive tokenization methods clearly outperform it. **Superpixel** segmentation underperforms other learned tokenization methods, indicating that bottom-up pixel grouping lacks the holistic understanding required to capture high-level semantic structures. **Object-level** tokenization based on panoptic segmentation shows strong in-domain performance (*e.g.,* on COCO and ADE-20K) but struggles in zero-shot settings, where the segmentation models have to generalize to unseen datasets. On PPP and PIN++, where all models are in *zero-shot* settings, our proposed **EPOC** demonstrates clear advantages over panoptic segmentation models and SAM ViT-B models, achieving Pareto optimal performance. EPOC also matches the performance of FastSAM and MobileSAMv2 and closely approaches SAM ViT-H models. Importantly, EPOC achieves this level of performance with a significantly smaller model size and offers substantially faster inference speeds, as will be presented soon.

**Token Monosemanticity Score.** As discussed in §2.2, one of the key goal of adaptive token segmentation is to minimize the occurrence of *polysemantic* tokens. We quantify this by calculating a token monosemanticity score, defined as the percentage of predicted token segments that *lie en-*

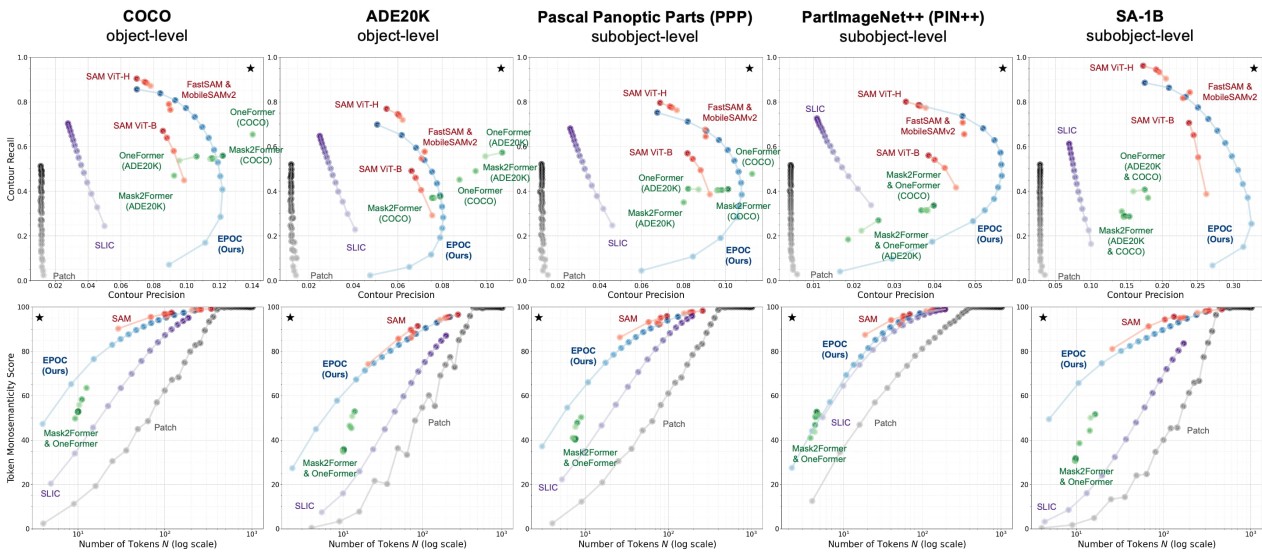

*Figure 4.* **Intrinsic evaluation of token segmentation**. Connected dots represent same model in different sizes or with different hyperparameters. **Top**: We measure the alignment between token segmentation and semantic annotations with boundary precision and recall. Our proposed EPOC achieves Pareto optimality compared to SAM ViT-B models and matches the performance of FastSAM and MobileSAMv2. **Bottom**: All subobject-level methods demonstrate clear advantages over object-level (in maximum achievable monosemanticity score) and static patch-based tokenization (in token efficiency).

*tirely within a single ground-truth region*, *i.e.,* no crossing of ground truth boundaries that separate semantically distinct regions (see Appendix B for more details). The bottom row in Fig. 4 presents the results, where token monosemanticity scores are plotted against the total number of tokens. For **patch-based** tokenization, smaller patches naturally avoid crossing ground-truth boundaries but result in significant token redundancy (large $N$). All *adaptive* methods outperform static patch, with **subobject-level** tokenization (SAM, EPOC, and SLIC) being able to approach high (*e.g.,* $> 90\%$) monosemanticity with varying levels of token efficiency, while **object-level** panoptic segmentation remains mostly below 60%. This observation supports our analysis in §3.1, where we hypothesized that the out-of-vocabulary problem poses a significant challenge for panoptic segmentation models.

**Computational Efficiency.** Image tokenizers must also offer fast inference. We measure throughput with a V100 (32GB) an 30 CPU cores by progressively spawning tokenizer processes (each with a batch size of 10) until a total of 30 processes. Table 1 shows the maximum achievable FPS and corresponding GPU utilization. With only 3.7M parameters, EPOC achieves up to 17.1 FPS with under 10% GPU utilization, outperforming all SAM-based methods, allowing minimal impact on VLM's computation. Profiling indicates `SegFormer`'s forward occupies only 5–7% of the total time, with CPU-based watershed taking the remainder. This explains why increasing threshold `t` can boost efficiency: the gap between seed regions become thin-

*Table 1.* Comparing computation efficiency of image tokenizers.

| Image Tokenizer | | Params (Millions) | Maximum FPS | GPU Usage% |
|---|---|---|---|---|
| Patch | | 0.0 | $+\infty$ | 0.0 |
| Mask2Former | `tiny` | 47.4 | 53.1 | 53.1 |
| | `large` | 215.5 | 34.9 | 44.6 |
| OneFormer | `tiny` | 50.7 | 17.3 | 99.1 |
| | `large` | 218.8 | 11.1 | 99.4 |
| SLIC Superpixel | `k=64` | | 15.4 | |
| | `k=128` | 0.0 | 13.2 | 0.0 |
| | `k=256` | | 11.0 | |
| SAM and Variants | `SAM ViT-B` | 93.7 | 0.6 | 96.5 |
| | `SAM ViT-H` | 641.1 | 0.4 | 92.4 |
| | `FastSAM` | 72.2 | 12.2 | 94.2 |
| | `MobileSAMv2` | 711.0 | 1.2 | 89.2 |
| **EPOC** (Ours) | `t=0.1` | | 13.4 | 5.7 |
| | `t=0.3` | 3.7 | 16.2 | 8.1 |
| | `t=0.5` | | 17.1 | 9.0 |

ner and the watershed step shortens. Optimized watershed implementations (Perret et al., 2019) could bring further improvements, but since our implementation already meets the throughput needs of training downstream VLMs, we leave further optimization to future work. More details are provided in Appendix B.

## 5. Extrinsic Evaluations

While our intrinsic evaluations assessed the innate properties of various image token segmentation methods, a complete picture requires extrinsic evaluations that measure their impact on downstream tasks. Compared to static patch,

adaptive image tokenization (§3) introduces new challenges of dynamic spatial arrangements and heterogeneous sizes and shapes. Traditional architecture cannot handle these properties, as they rely on simplified design that assumes predetermined grid structures and static raster-order patch arrangement. In this section, we first detail our methodology of incorporating such information into visual tokens in §5.1, then present the setup and results in §5.2 and §5.3.

## 5.1. Token Embedding for Adaptive Segmentation

Given an image $\mathbf{X}$ and token segmentation map $\mathbf{M}$, the goal is to generate visual tokens $\{\mathbf{x}_i\}_{i=1}^N$, where each $\mathbf{x}_i$ corresponds to one segment in $\mathbf{M}$. We first encode the content and position of each token into $\mathbf{x}_i^c$ and $\mathbf{x}_i^p$, respectively representing "*what it is*" and "*where it is*", and fuse them into visual tokens $\mathbf{x}_i$, which are then fed to $f$ to learn $\mathbf{X} \mapsto \mathbf{Y}$.

**Content Embedding.** We allow a vision encoder to extract a feature map from the input image, *i.e.*, $\texttt{feature}(\mathbf{X}) \in \mathbb{R}^{H_e \times W_e \times C}$, where $H_e$ and $W_e$ are spatial dimensions, and $C$ represent channels. For the case where $\texttt{feature}(\cdot)$ is an identity function, it allows $f$ to directly learn from raw pixels. Despite such special case, the spatial resolution of $\texttt{feature}(\mathbf{X})$ is usually lower than $\mathbf{M}$ whose resolution is $H \times W$, so we $\texttt{upsample}$ the feature map to match the resolution of the mask $\mathbf{M}$ via $\mathbf{X}' \coloneqq \texttt{upsample}(\texttt{feature}(\mathbf{X})) \in \mathbb{R}^{H \times W \times C}$.

Given $\mathbf{X}'$ and $\mathbf{M}$, we aggregate the pixel features corresponding to each segment to obtain the content embedding. Specifically, we define $\mathbf{x}_i^c \coloneqq \texttt{pool}\big(\{\mathbf{X}'[h, w] \mid \mathbf{M}[h, w] = i\}\big)$, where $\mathbf{X}'[h, w] \in \mathbb{R}^C$ is the feature vector at pixel coordinates $(h, w)$ and $\texttt{pool}(\cdot)$ is a pooling function (*e.g.*, average pooling). Although more sophisticated strategies (*e.g.*, flattening ROI features or learning a Perceiver Resampler (Alayrac et al., 2022)) may further refine these embeddings, we find that simple average pooling is effective if the segments are reasonably monosemantic.

**Position Embedding.** Adaptive tokenization introduces dynamic spatial arrangements and irregular token shapes, requiring position embeddings to preserve such information. Each segment in $\mathbf{M}$ has that information, but directly encoding each mask at the full image resolution is costly. Instead, we exploit the prior that most segments form connected components, which allows a more compact representation via box-mask decomposition.

Specifically, we compress the raw mask into **token location** (represented as a bounding box), and **token shape** (a cropped mask inside the bounding box). For each token $i$, we compute its bounding box $b_i \in [0, 1]^4$ using the normalized $\texttt{<xywh>}$ format, and we also extract a cropped binary mask $\mathbf{m}_i \in \{0, 1\}^{H_m \times W_m}$ that indicates the exact shape of the segment (with $H_m < H$ and $W_m < W$). We then

define the position embedding as $\mathbf{x}_i^p \coloneqq \texttt{encode}(\mathbf{m}_i) \oplus b_i$, where $\oplus$ denotes concatenation. The $\texttt{encode}(\cdot)$ function converts each cropped mask into a vector, and we implement it with downsampling and flattening. More sophisticated encodings, such as CNN-based encoders, can be incorporated. However, as the amount of computation is dependent on the number of tokens $n$, we keep it simple in this paper.

**Fusion.** Finally, the content embedding $\mathbf{x}_i^c$ and the position embedding $\mathbf{x}_i^p$ are fused via a small MLP: $\mathbf{x}_i \coloneqq \text{MLP}\big(\mathbf{x}_i^c \oplus \mathbf{x}_i^p\big)$, yielding the $i$-th visual token. Collectively, these tokens $[\mathbf{x}_1, \mathbf{x}_2, \dots, \mathbf{x}_N]$ serve as the input to the downstream model for tasks such as image classification or caption generation.

## 5.2. Evaluation Setup

**VLMs.** We follow the strategy outlined in §5.1 to transform images into visual tokens, then feed them to text-only LLMs with a Llama-style architecture (Dubey et al., 2024). We apply the standard next-token prediction loss only on text tokens, ignoring the loss for the visual tokens. We train the entire LLM and the MLP projection while freezing the feature embedding (Appendix C provides more details on our evaluation setup).

**Image Tokenizer.** Due to the high cost of training VLMs, we only select a set of efficient representative token segmentation approaches to compare, which include patch tokenization, object-level panoptic segmentation (Mask2Former $\texttt{tiny}$ and $\texttt{large}$ trained on COCO), and subobject-level SLIC and EPOC. We use varying patch's p, SLIC's k and EPOC's t to sweep over different granularity.

**Datasets.** We train VLMs on four datasets: **ImageNet-1K** (Deng et al., 2009): A object recognition dataset widely-used in CV. We treat class names as text target. **ShareGPT-4V** (Chen et al., 2024): A detailed captioning dataset commonly used for vision-language alignment. **Pixmo-cap** (Deitke et al., 2024): A high-quality dataset featuring rich human-annotated dense captions. **CLEVR-cap**: A new dataset introduced for fast ablation experiments. The dataset is generated by using CLEVR (Johnson et al., 2017) metadata to create detailed captions that enumerate the attributes of each object from left to right. Captions follow the template of "A total of {n} objects: a {color} {size} {material} {shape}, ...". CLEVR-cap involves core visual understanding capabilities, including counting, spatial reasoning, and recognition of object color, size, material, and shape. By constraining to a clean synthetic domain, it significantly facilitates evaluation cycles while maintaining experimental rigor.

## 5.3. Results and Discussions

**Adaptive Tokenization Facilitates Image Understanding.** Figure 5 presents validation perplexities of VLMs trained

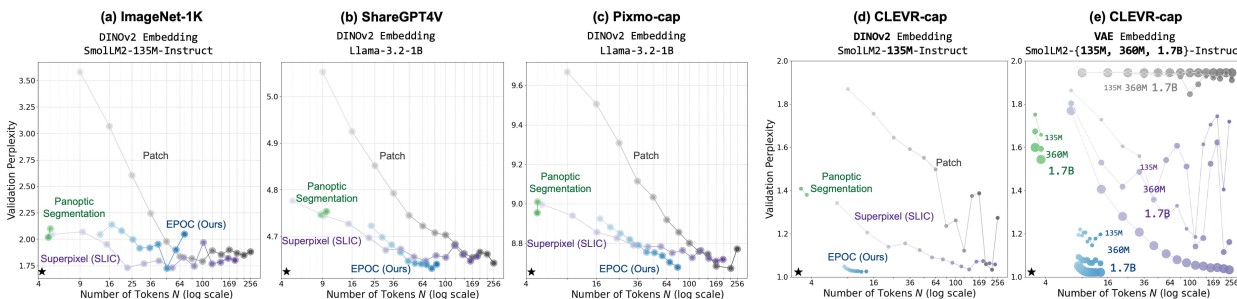

*Figure 5.* **Extrinsic evaluation of token segmentation**. **(a-d)**: Adaptive token segmentation shows clear advantage over patch tokenization, with subobject-level SLIC and EPOC being able to approach lower perplexity than object-level ones. **(e)**: when using VAE embedding which is less semantically expressive, patch-based model failed to converge, while adaptive tokenizers works fine.

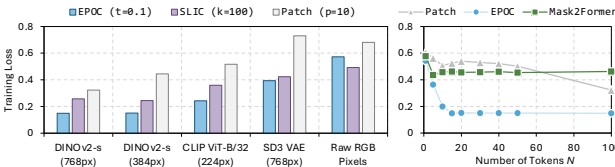

*Figure 6.* **Left**: adaptive subobject-level tokenization outperform patch tokenization consistently across different embeddings. **Right**: EPOC-based tokenization is robust to token truncation, showing the potential of further improvement in token reduction.

different tokenizers (each point corresponds to an individual run). Across all datasets, most *adaptive* methods lie to the left or below the patch baseline, particularly in low $N$ regime, indicating they can either reduce the number of tokens for similar performance or improve generalization under comparable token budgets. Among them, subobject-level EPOC and SLIC stand out for matching or surpassing the best performance of very small patches while using much fewer tokens. In contrast, object-level tokenization is more efficient than large patches but still lags subobject-level approaches in minimal achievable perplexity.

**Compatibility to Different Token Embeedings.** Figure 5 also compares the DINOv2 (Oquab et al., 2024) with VAE embeddings on CLEVR-cap. Notably, patch-based tokenization struggles to converge when using the weaker VAE features, even with an LLM scaled to 1.7B parameters. In contrast, subobject-based models remain effective, demonstrating that adaptive tokenization can simplify the learning of image understanding. Additionally, Figure 6 (left) presents an ablation study across various token embeddings on CLEVR-cap using three token segmentation methods that produce similar $N$. Subobject tokenization consistently outperforms static patch tokenization while maintaining a comparable number of visual tokens. Interestingly, CLIP (Radford et al., 2021) underperforms DINOv2 in our setup, likely due to its lack of dense supervision and the lower feature-map resolution of only $7\times7$.

**Robustness to Token Reduction.** We explore the potential of further reducing visual tokens by truncation. We do random drop-out for patch tokenizer, while for object- and subobject-level tokenization, we start dropping from the smaller ones. The result in Figure 6 (right) shows that adaptive tokenization methods exhibit stronger robustness compared to patch tokenization. Their long-tail token size distribution (Appendix B) ensures that discarding the smallest tokens has minimal impact on performance, whereas random patch dropout leads to notable degradation. This suggests that adaptive approaches hold even greater potential for optimizing token efficiency.

## 6. Related Work

Tokenization, a critical step in the NLP (Gastaldi et al., 2024), has been historically extensively studied (Mielke et al., 2021). Despite active explorations (Pagnoni et al., 2024; team et al., 2024), subword remains the predominant choice for language modeling. Over the years, the effectiveness of subwords has been analyzed from multiple perspectives, ranging from intuitive explanations about reducing rare words (Sennrich et al., 2016; Liu et al., 2019) to information-theoretic measurements (Zouhar et al., 2023; Schmidt et al., 2024; Goldman et al., 2024). The commonality among these analysis is the idea of a good tokenizer should maintain a balanced distribution over a manageable size of vocabulary, while providing reasonable and token sequence length. The principle of balanced distribution parallels our goal of preventing polysemantic tokens with limited appearance frequency—akin to rare words.

The idea of adding adaptivity to visual tokenization has also been explored in previous studies. Some methods (Lew et al., 2024; Aasan et al., 2024; Ke et al., 2022) rely on bottom-up superpixel grouping, which lacks semantic awareness. Feature-based methods, such as slot attention (Locatello et al., 2020) and various token pooling methods (Haurum et al., 2023; Ronen et al., 2023; Marin et al., 2023;

Feng & Zhang, 2023; Huang et al., 2022), suffer from low-resolution feature maps which limits fine-grained segmentation, and unreliable segmentation quality in early training stages. Our approach of using a separate segmentation model, EPOC, avoids these issues, and outperform SAM-based segmentation (Kim et al., 2024) in terms of efficiency. Additionally, there are also works on image tokenization focusing on improving token *embeddings* for understanding or generation (Ramesh et al., 2021; Hansen-Estruch et al., 2025), while leaving token *segmentation* simply as patch grids. Our work can be combined with these efforts.

## 7. Conclusion

In this work, we introduce the concept of subobject image tokenization and propose EPOC, an efficient and panoptic image tokenizer that combines boundary detection and watershed segmentation to achieve high-quality, monosemantic token segmentation while maintaining computational efficiency. Through extensive evaluations, we demonstrate that EPOC not only aligns well with human annotations of visual morphology, achieves better token monosemanticity, but also enabling faster convergence, better generalization of VLMs. Our findings highlight the potential of adaptive segmentation in improving vision models.

## Limitations

Despite the advantages of EPOC and adaptive (especially subobject-level) tokenization, there are several limitations and areas for future improvement. Below, we outline several key considerations:

- Our boundary prediction model is based on SegFormer, which produces feature maps at a quarter of the input resolution. Consequently, very fine or thin structures (*e.g.,* strands of hair or texts in small font) may not be segmented precisely. Using different architecture and with higher resolution output can be a potential way to alleviate this issue.

- Our extrinsic evaluations employ LLMs with a maximum of 1.7 Billion parameters. While this scale is already substantial for vision backbones (compared to ViT), it remains limited compared to the largest LLMs available.

- On the other hand, although EPOC is lightweight and fast, it still adds a separate segmentation process. This overhead may be less critical for computationally heavy VLM training or inference pipelines but warrants attention in latency-sensitive applications. We believe, however, that this overhead is more than offset by efficiency gains—particularly in high-dimensional data such as video (Zheng et al., 2025)—where token number explosion is a persistent bottleneck (and mod-

ern VLMs are still using a relative low FPS rate as a consequence).

- Adaptive tokenization can be interpreted as a learned compression scheme, grouping correlated pixels to reduce token count. While it has been shown beneficial for VLM, it inevitably discards intra-token spatial structure, which might be valuable in settings that demand ultra-fine resolution or pixel-level edits. This issue is similar to the disadvantage of subword-based LLMs on character-level task. Adding more adaptivity to adaptive tokenization (*i.e.,* task conditioned segmentation) can be potential ways to address this.

## Acknoledgement

We sincerely thank Quentin Garrido, Yejin Bang, and Willy Chung for their valuable discussions, insightful feedback, and constructive comments on this work. Their input has significantly contributed to improving the clarity and quality of this paper. Samuel Cahyawijaya, Jianfeng Liu and Baoyuan Wang contributed to this work before Delong Chen and Pascale Fung joined FAIR at Meta.

## Impact Statement

This paper presents work whose goal is to advance the field of Machine Learning. There are many potential societal consequences of our work, none which we feel must be specifically highlighted here.

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

# Appendix

## A. Token Segmentation Models

**Patch-based Tokenization.** We divide the image into non-overlapping $p \times p$ patches. The p ranges from 2 to 32 in intrinsic evaluations (§4) and ranges from 9 to 16 in extrinsic evaluations (§5).

**Panoptic Segmentation.** We employ Mask2Former and OneFormer, two popular types panoptic segmentation models. Our implementation is based on their official releases in Huggingface, which include models with different sizes and models trained on both COCO and ADE20K:

- `facebook/mask2former-swin-tiny-coco-panoptic`
- `facebook/mask2former-swin-small-coco-panoptic`
- `facebook/mask2former-swin-base-coco-panoptic`
- `facebook/mask2former-swin-large-coco-panoptic`
- `facebook/mask2former-swin-large-ade-panoptic`
- `shi-labs/oneformer_ade20k_swin_tiny`
- `shi-labs/oneformer_ade20k_swin_large`
- `shi-labs/oneformer_coco_swin_large`

**Superpixel Segmentation.** We vary the k in $\{2^2, 3^2, 4^2, ... 16^2\}$ for $k$-means clustering, effectively controlling the number of superpixels (Fig.7). We use the SLIC implementation in `scikit-image` library (url), where the k corresponds to the `n_segments` parameter. Other hyperparameters are kept as default.

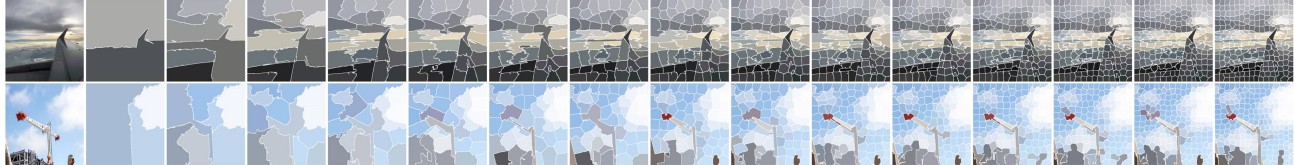

*Figure 7.* SLIC superpixel segmentation results with different k.

**SAM, FastSAM, MobileSAMv2.** Token segmentation is generated by prompting the SAM on a regular grid of points and merge the resulting masks using the official "automatic mask generation" implementation. We test SAM backbones with different sizes: `ViT-B` and `ViT-H`. To explore the limit of fine-grained segment everything, we further increase the prompt density from the default $32 \times 32$ to $48 \times 48$ and $64 \times 64$ and include an one-layer multi-scale augmentation (MS) based on the `ViT-H` model. One the other hand, we scale down the density to $24 \times 24$, $16 \times 16$ and $8 \times 8$ based on the smallest `ViT-B` model for more conservative and efficient segmentation. We also include two efficient variants of SAM: **FastSAM** (github url) which uses lightweight CNN detector instead of ViT, and **MobileSAMv2** (github url) which replaces grid point prompt with a learned proposal, reducing the number of forward pass through the mask decoder. Fig. 8 compares their segmentation results on SA-1B samples.

**Proposed EPOC** For boundary detection, we trained a `SegFormer-b0` model on SA-1B dataset for 2 epochs, where mask annotations are converted into binary boundary maps as the target. Specifically, for every segmentation in an image, we computed the difference between its dilation and erosion using a circular kernel with the size of 5, and stacked them together as the boundary label.

The training was performed on a single NVIDIA 8×A100 machine. We used a effective batch size of 64, learning rate of 1e-4 with a constant schedule and 5000 warmup steps. The model was optimized using AdamW with a weight decay of 0.05. Images were resized to 1024×1024, and the loss was computed as a pixel-wise binary cross-entropy. During training, we applied random horizontal flipping and color jittering as data augmentation. Fig. 9 visualizes the boundary probability maps predicted by the trained model on unseen Pixmo-cap samples with 1024px input resolution. We apply watershed algorithm

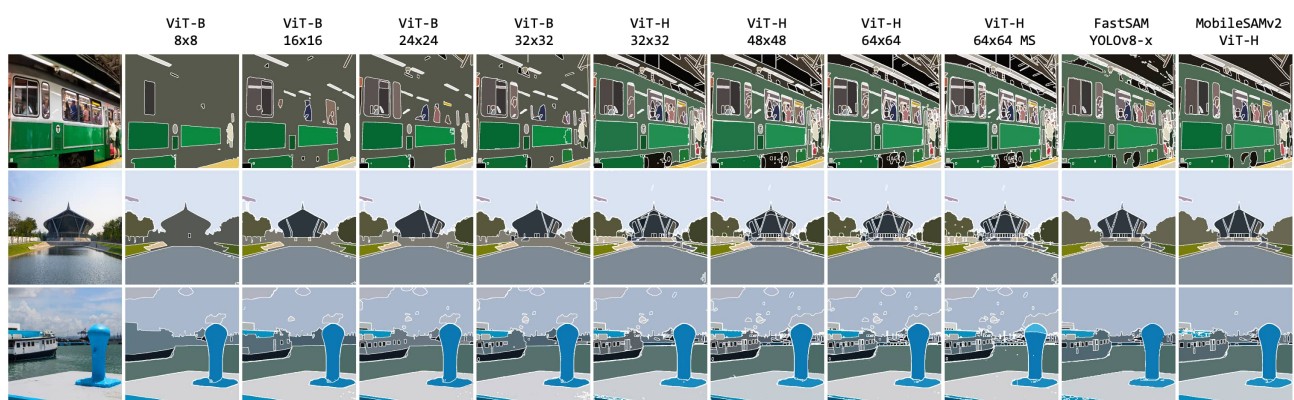

*Figure 8.* "Segment Everything" results from different SAM models.

(with `scikit-image`) on sigmoid normalized boundary probability map to get panoptic segmentation. Fig. 10 visualizes how different threshold `t` affects the segmentation granularity.

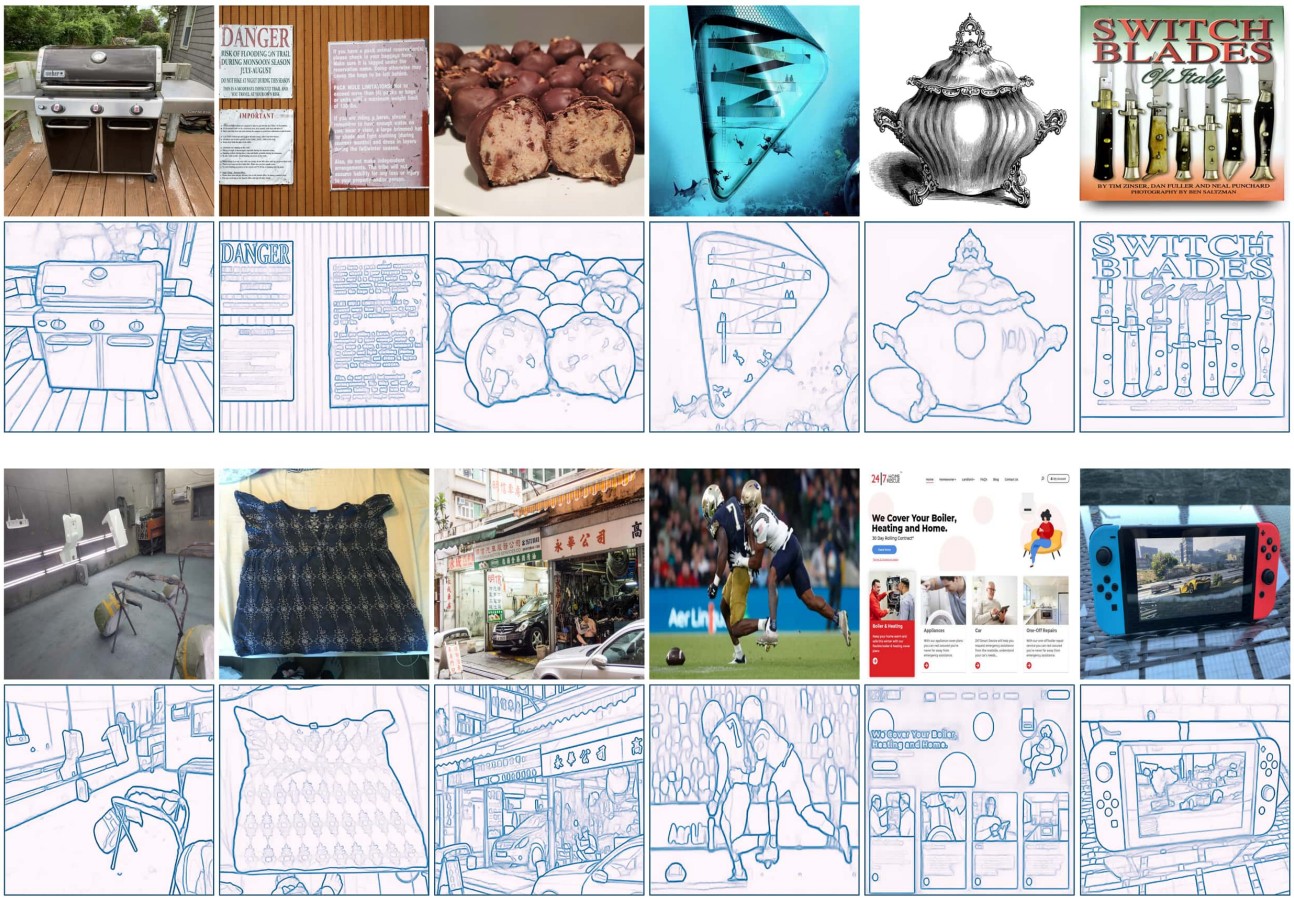

*Figure 9.* Visualization of predicted boundary probability maps on Pixmo-cap samples.

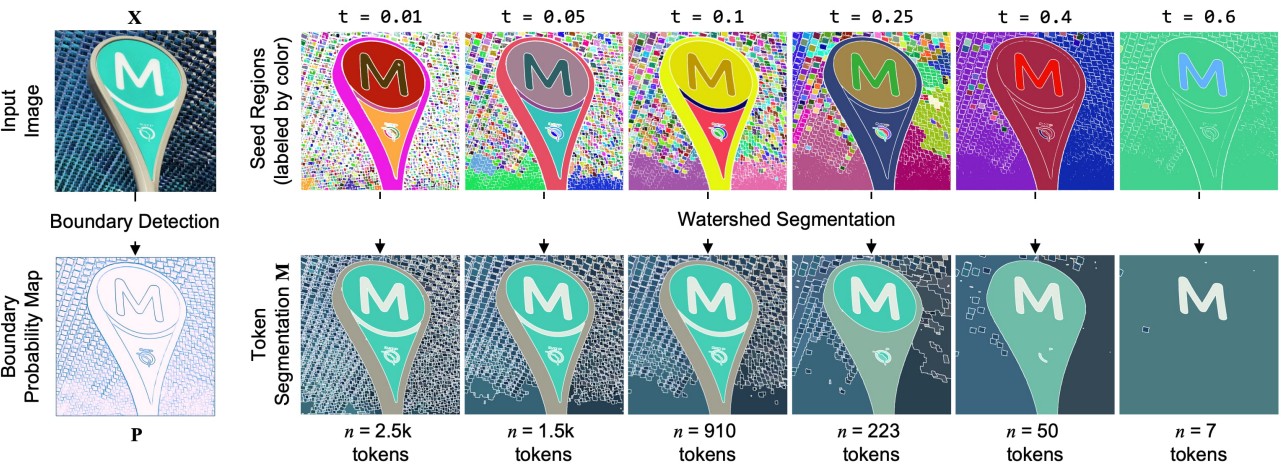

*Figure 10.* The threshold `t` provides flexible control over segmentation granularity: lower `t` produces more seed regions and thus finer segmentation, while higher `t` results in fewer, larger tokens.

## B. Intrinsic Evaluations (Extended)

**Datasets.** For COCO, PPP, PIN++, and SA1B, we randomly sample 3k images for efficient evaluation. For ADE-20K, we include all 2k samples in the validation set. As we employ boundary-based metrics for evaluation, we convert their mask annotations into boundaries by applying morphological dilation and erosion operations with a kernel size of 3 on the mask, and then taking the difference to isolate the boundary. Fig. 11 provide extended visualization of dataset examples.

**Boundary Precision and Recall.** A small $\text{tolerance}_{\text{recall}} = 5\text{px}$ is applied to the predicted boundaries to compensate for slight pixel misalignments when calculating recall.

**Token Monosemanticity Score.** A token is considered monosemantic if it does not include pixels from more than one ground-truth semantic region. Formally, let $\mathbf{X} \in \mathbb{R}^{H \times W \times 3}$ be an image, $\mathbf{M} \in \{0, \dots, N-1\}^{H \times W}$ the predicted token map, and $\mathbf{M}^* \in \{0, \dots, K-1\}^{H \times W}$ the ground-truth. The $i$-th predicted token is $\mathcal{T}_i = \{(h, w) \mid \mathbf{M}[h, w] = i\}$.

Define the indicator:

$$\mathbb{I}_{\text{mono}}(\mathcal{T}_i) = \begin{cases} 1, & \exists k \text{ s.t. } \forall (h, w) \in \mathcal{T}_i, \ \mathbf{M}^*[h, w] = k \\ 0, & \text{otherwise.} \end{cases}$$

Then the monosemanticity score is defined as:

$$\text{Mono}(\mathbf{M}, \mathbf{M}^*) = \frac{1}{N} \sum_{i=0}^{N-1} \mathbb{I}_{\text{mono}}(\mathcal{T}_i).$$

This class-agnostic score quantifies how often a token is wholly contained in a single semantic region, capturing the absence of polysemantic tokens. To measure this: 1) we erode each predicted token segment by $\text{tolerance}_{\text{monosemanticity}} = 25\text{px}$ so that only the "core" of each token remains; 2) Tokens whose eroded region does not intersect a ground-truth boundary are deemed monosemantic. This captures whether a token merges multiple semantics into a single segment.

**Token Size Distribution.** To better understand how each segmentation method allocates tokens, we analyze the relative area of each token by sorting them from largest to smallest. Figure 12 plots the average token size as a percentage of the total image area on a log scale, with the $x$-axis denoting the rank of the token in descending order of size. **Patch-based segmentation** yields a *uniform* token size distribution. In **panoptic segmentation**, only a few tokens occupy a large portion

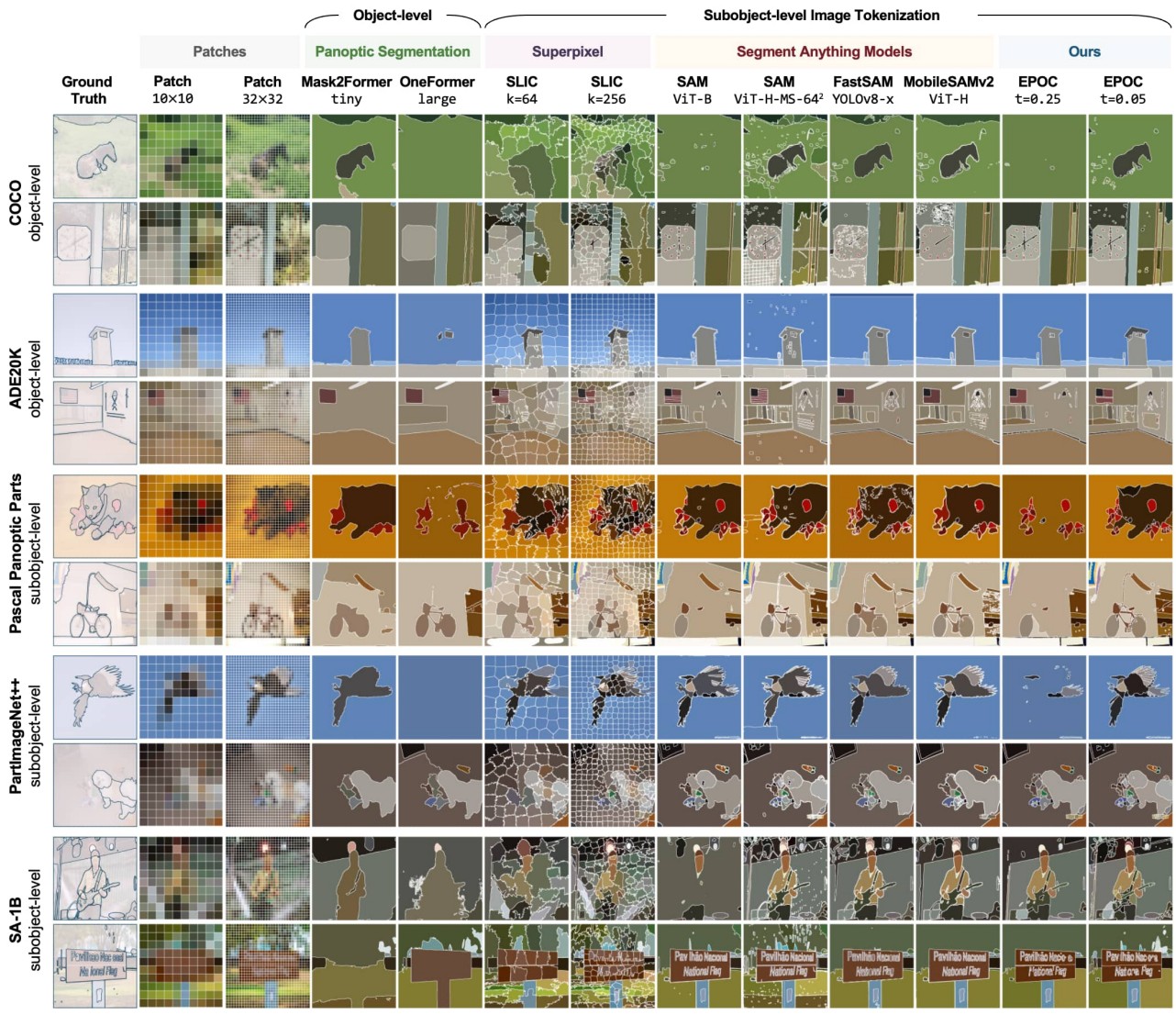

*Figure 11.* Intrinsic evaluation dataset examples and token segmentation results.

of the image. **EPOC** exhibit a pronounced *long-tailed* distribution. Those observations can explain our findings in the token truncation experiments (§5.3).

**Computational Efficiency.** We measure throughput on an NVIDIA V100 (32GB) with 30 CPU cores by gradually increasing the number of parallel processes, each running one tokenizer. During these multi-process runs, we record wall-clock time, GPU memory consumption, GPU utilization, and any out-of-memory events. This setup reveals how well each tokenizer scales in a concurrent inference setting and highlights potential memory bottlenecks when more processes are added. Fig. 13 visualize the dynamics of FPS and average GPU utilization with increasing number of process.

## C. Extrinsic Evaluations (Extended)

**Datasets.** Fig. 14 provide visualization of dataset examples. ImageNet-1k and CLEVR provide official validation splits, we use 5k samples from them for efficiency. For Pixmo-cap, we randomly sample 5k samples as validation, and for ShareGPT-4v, we treat 5k samples randomly selected from the GPT-4V generated captions as validation split.

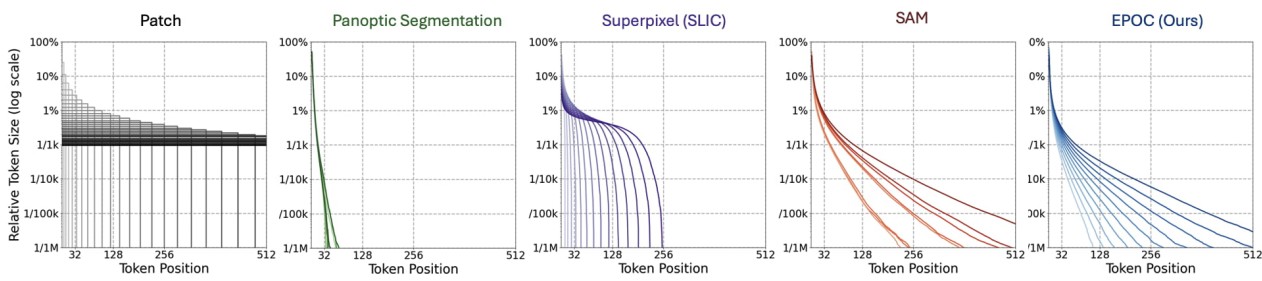

*Figure 12.* Visualization of token size distribution.

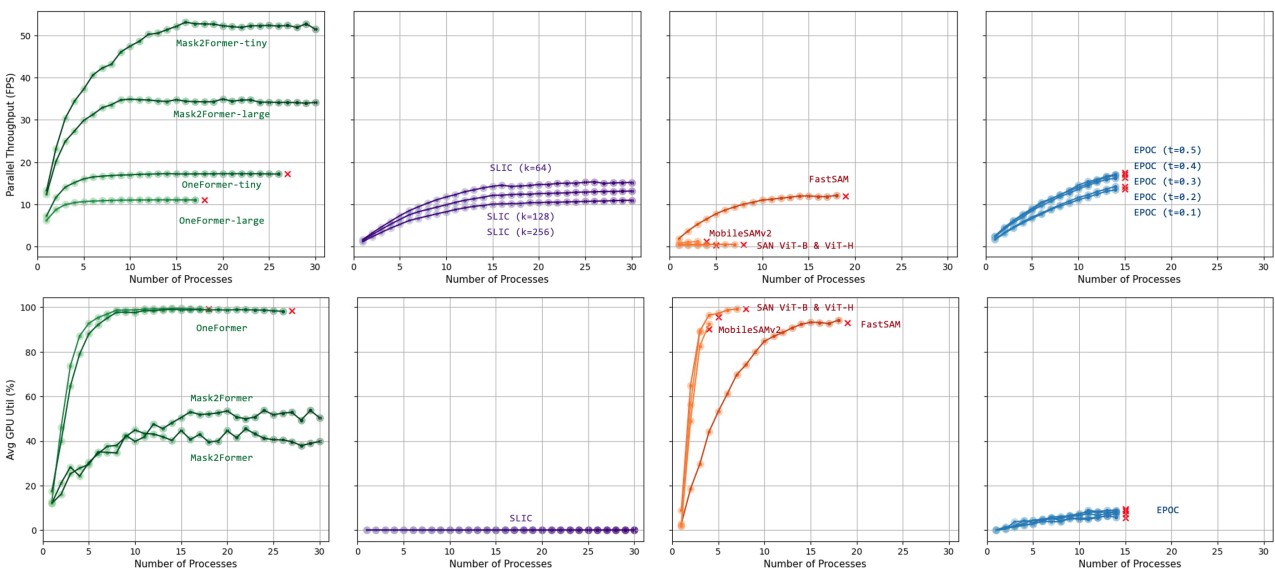

*Figure 13.* Detailed computational efficiency evaluation results.

**VLM Architecture and Training**    We use a two-layer MLP as the connector between embeddings and LLM. The width is $\times 4$ of LLM's hidden state dimension. We freeze the image feature extractor and do end-to-end fine-tune the small MLP projection plus the LLM. For CLEVR-cap, ImageNet-1k, ShareGPT4V, and Pixmo-cap datasets, we respectively train the model for 30, 1, 1, 3 epochs, with a batch size of 512, 256, 256, and 256. Max tokens are set to 100 for EPOC and 64 for Mask2Former tokenizer. We use AdamW with learning rate $1 \times 10^{-4}$, cosine decay or constant scheduling, and 500 warmup steps. Mixed-precision (bf16) is used to accelerate training. We do standard language-modeling next-token prediction on text tokens only.

**Convergence Speed**    We measure convergence speed using average training loss, which effectively shows how quickly the model fits the training data. Fig. 15 (top) demonstrates that the convergence speed correlate well with generalization performance (bottom), both showing the advantage of subobject-level image tokenization.

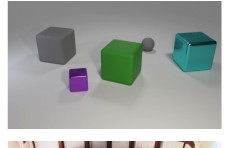

**CLEVR-cap (70k samples)**

Total 5 objects: a large gray rubber cube, a small purple metal cube, a large green rubber cube, a small gray rubber sphere, a large cyan metal cube.

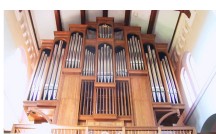

**ImageNet-1k (1.28M samples)**

organ

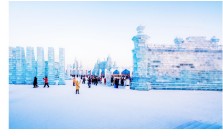

**ShareGPT4v (1.24M samples)**

The image captures the breathtaking view of the Harbin Ice and Snow World in China. The perspective is from a low angle, looking up at the towering ice sculptures that dominate the scene. These sculptures, meticulously crafted from blocks of ice, take the form of buildings and other structures, their intricate details a testament to the skill and creativity of the artists. The colors in the image are predominantly blue and white, reflecting the cool tones of the ice and snow. The sky above is a clear blue, providing a striking contrast to the icy landscape below. The image is taken during the day, and the bright sunlight illuminates the sculptures, casting long shadows and highlighting the texture of the ice. The overall effect is one of awe and admiration for the artistry and skill involved in creating such a spectacle.

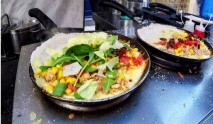

**Pixmo-Cap (707k samples)**

In this industrial-style kitchen, a bustling scene features a large silver gallon pot in the background and a stainless steel cooking station in the foreground. Center stage are two sizable pans with black metal handles and stainless steel bottoms, both brimming with a colorful array of ingredients. The pan on the left appears laden with spinach leaves, arugula, chunky yellow pieces that resemble potato or corn, possibly some kind of meat, and vibrant red tomatoes or tomato chunks. Meanwhile, the pan on the right showcases yellow bits of corn, red tomatoes, fiery red chilies, and some ground meat. A messy stovetop suggests a lively cooking session, with crumbs scattered and two sauce bottles—one red and one clear—positioned to the far right. Overhead, stainless steel shelves and pots add to the industrious ambiance of the kitchen. This vibrant scene captures the making of a hearty breakfast, potentially a scrambled mixture or an omelet, full of fresh, colorful ingredients sizzling together.

*Figure 14.* Examples from extrinsic evaluation datasets that are used to train VLMs.

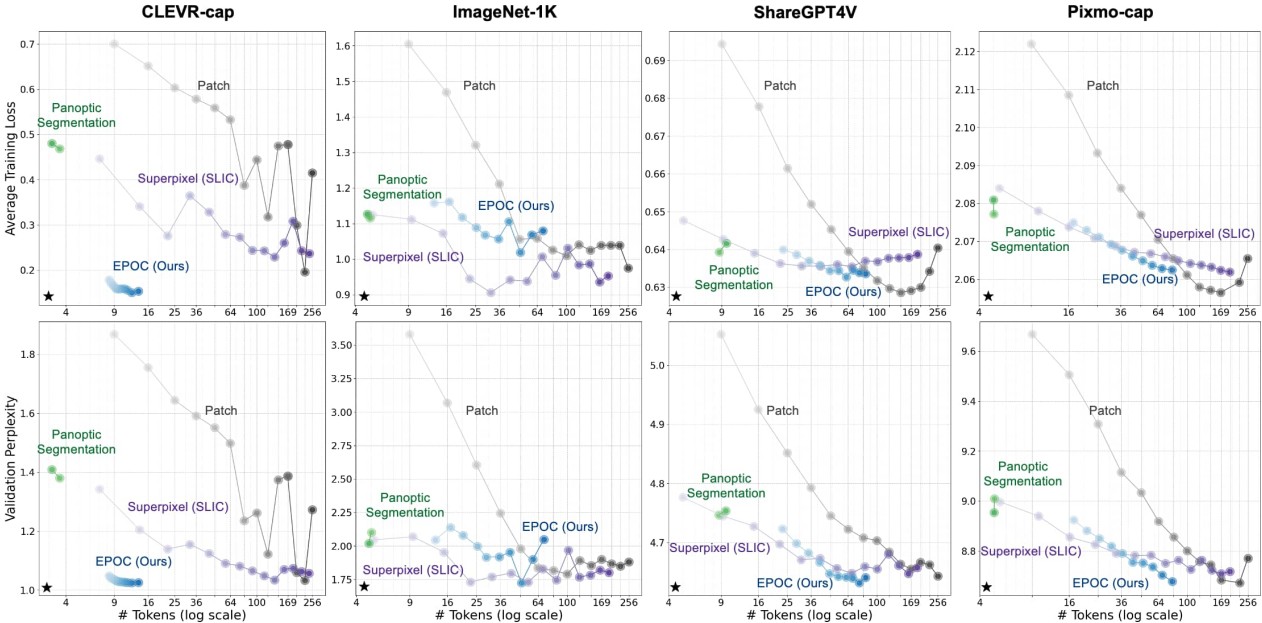

*Figure 15.* Comparing average training loss (measuring convergence speed) and validation perplexity.

