# OpenReview forum: "Subobject-level Image Tokenization"
_ICML.cc/2025/Conference — ICML 2025 poster_

### Official Review · Reviewer_D5bb · 2025-03-10

**Overall Recommendation:** 2

**Summary:**

This paper presents a method to encode an image at sub-object level. Specifically, it first detects edges and boundaries in the image with a small model, then utilizes the watershed algorithm to segment the image into sub-object parts. The authors conduct both intrinsic evaluations to validate the segment quality and extrinsic evaluations to verify the derived sub-object token embeddings.

## update after rebuttal
Thank the authors for the detailed response. However, the rebuttal did not satisfactorily address my concern regarding the lack of VQA evaluation. Given that the primary objective of token segmentation is to improve token embeddings for visual understanding, it is crucial to assess performance on diverse understanding-oriented VQA benchmarks, not just simple image captioning tasks. I am NOT requesting that the authors develop a SOTA VLM, but rather that they provide a fair comparison with CLIP embeddings that do not involve token segmentation on VQA benchmarks. The authors’ reluctance to provide such an evaluation leads me to question the true effectiveness of the proposed method. Therefore, I will maintain my rating as weak reject.

**Claims And Evidence:**

The claims in this paper supported by experimental results or prior studies.

**Essential References Not Discussed:**

N/A

**Experimental Designs Or Analyses:**

The experimental designs are solid and fair.

**Methods And Evaluation Criteria:**

The extrinsic evaluation benchmarks used in the paper are not common for the VLM evaluation.

**Other Comments Or Suggestions:**

I suggest that the authors move the introduction of 'Token Embedding for Adaptive Segmentation' (Section 5.1) to the 'Method' section (Section 3), as deriving sub-object token embeddings should be an indivisible part of the proposed method.

**Other Strengths And Weaknesses:**

**Strengths:**
- The idea of tokenizing an image at sub-object level is intuitive and intriguing. It reduces the number of tokens required to encode an image compared to traditional patch-wise tokenization, and facilitates fast convergency in downstream tasks.
- The paper is well-written and easy to follow.

**Weaknesses:**
- Sub-object image tokenization involves two stages: first, partitioning an image into sub-object parts, and second, encoding each part into tokens. This paper overlooks the design of the latter stage. It still relies on a patch-wise encoder to generate 2D feature maps, and then pool the features corresponding to each segment. This design could fail to fulfill the monosemanticity of sub-object tokens.
- The extrinsic evaluation benchmarks used in this paper are not common in VLM evaluation. The authors are encouraged to conduct evaluation under standard settings, i.e., the LLaVA framework.

**Questions For Authors:**

How does the throughput of the sub-object tokenizer compare to that of the patch-wise tokenizer (e.g., CLIP or DINOv2) for a similar model size, including the phase of deriving token embeddings?

**Relation To Broader Scientific Literature:**

This paper studies the image tokenization problem from a novel perspective, i.e., at sub-object level, which is analogous to sub-word tokenization in NLP.

**Theoretical Claims:**

There is no proof or theoretical claim in this paper.

---

> ### Author Rebuttal · Authors · 2025-04-01
>
> Thank you for your constructive feedback! We realize that there are some concerns that arise from the interplay between token segmentation and token embedding. Therefore, before addressing your comments point-by-point, we first clarify how and why we disentangle these two components in our paper:
>
> In this study, we explicitly separate the image tokenization into two stages: 1. **token segmentation** and 2. **token embedding**. Our paper emphasizes the segmentation step, which has been largely overlooked in existing research focusing primarily on improving embeddings. **Our claim** is that: **adaptive segmentation** facilitates better image understanding, and this is consistent across **different embedding** methods.
>
> We adopted this separation due to several considerations:
>
> - Segmentation and embedding have fundamentally different roles and characteristics. They are agnostic to each other and can be flexibly combined.
>
> - Disentangling these two components allows controlled experiments and clearer attribution of improvements.
>
> - This separation aligns well with previous works investigating token segmentation methodologies, both in language modeling and image understanding literature.
>
> Below, we provide detailed responses addressing each of your specific comments.
>
> ---
>
> ### 1. **Token Embedding for Subobject Segmentation** (Weakness 1)
>
> > **TL;DR** Rather than overlooking token embeddings, our work demonstrates that adaptive segmentation consistently enhances image understanding **across various embedding methods**.
>
> Existing studies typically focus on improving token embedding quality while defaulting to simple, uniform patch-based segmentation. In contrast, our paper explicitly addresses the overlooked aspect of adaptive token segmentation. This segmentation method is designed to remain **agnostic** of embedding choices, facilitating a controlled evaluation of segmentation effectiveness across diverse embedding schemes, as in Fig.6 (left).
>
> Although popular ViT-based encoders are convenient and commonly used, they are by no means mandatory for our method. Indeed, as in Fig. 5(e), our EPOC achieves **stronger advantages** when coupled with **convolutional VAE** encoders or using raw **RGB pixels**.
>
> ---
>
> ### 2. **Extrinsic Evaluation Settings** (Weakness 2)
>
> > **TL;DR**  We indeed largely adopt the LLaVA framework, but with some minimal sufficient modifications. Without these changes, the original LLaVA framework **cannot provide meaningful evaluation** for adaptive token segmentation methods.
>
> The goal of this extrinsic evaluation is to probe the effectiveness of adaptive segmentation, not to propose any new VLM framework. The VLM evaluation setup in our paper closely follows common practices already established in the literature. The commonalities includes:
>
> - We adopt the standard "Visual Encoder → MLP Connector → LLM" architecture from LLaVA.
>
> - We adopt visual encoders (CLIP, DINOv2, VAE), MLP settings (2-layers), and LLM (autoregressive decoder-only Transformer) that is well aligned with existing studies.
>
> - We adopted ShareGPT4V and Pixmo-cap, which are both widely used for pretraining VLMs.
>
> However, minimal changes were necessary:
>
> - Standard LLaVA assumes fixed-size and raster-ordered patches, thus omitting positional embeddings. We introduced positional embeddings specifically to handle adaptive tokenization, which involves irregular segment shapes and arrangements.
>
> - Standard VLM benchmarks commonly evaluate visual reasoning through tasks like VQA, reflecting both vision and LLM capabilities. Since our goal is exclusively to measure improvements in image understanding capability, we only measure the caption quality—thereby minimizing confounding from LLM knowledge and reasoning factors.
>
> ---
>
> ### 3. **Throughput Comparison** (Questions For Authors)
>
> According to the disentanglement between token segmentation and token embedding, we interpret *“patch-wise tokenizer”* in the question as combining patch segmentation with CLIP or DINOv2 embeddings. A fair throughput comparison thus requires combining the same embedding backbones with subobejct segmentation. Since **embedding methods do not impact segmentation latency**, the throughput conclusions from Table 1 directly apply – *although patch segmentation is negligible in latency, the overhead introduced by our EPOC is minimal and practically insignificant relative to overall VLM training*.
>
> ---
>
> ### 4. **Structure and Clarity** (Other Comments Or Suggestions)
>
> Thank you for this constructive suggestion. While our original intention was to clearly separate the token segmentation from embedding strategies for conceptual clarity, we acknowledge that they are methodologically relevant. We will reorganize our paper in the final version to move "Token Embedding for Adaptive Segmentation" (Section 5.1) into the "Method" section (Section 3) as suggested.
>
> ---
>
> Once again, we sincerely appreciate your detailed review and valuable suggestions!

---

> > ### Comment · Reviewer_D5bb · 2025-04-02
> >
> > Thanks for the detailed responses, which address some of my concerns. However, I disagree that token segmentation and token embedding should be disentangled, because in this case, token embedding still suffers from fused multiple semantics within a single token (polysemanticity), an undesirable property that hinders effective learning of token representations, as mentioned in the introduction. Applying token segmentation on top of the fused token embedding is less effective than deriving segmented token embedding in a bottom-up manner.
> >
> > Besides, the author's response did not convince me for not using VQA benchmarks for evaluation. As suggested in Cambrian-1, combining VLM frameworks with VQA benchmarks is considered a modern evaluation protocol for visual representations.

---

> > > ### Author Response · Authors · 2025-04-03
> > >
> > > Thank you for the response and thoughtful follow-up comments. Below, let us clarify the points with additional evidence:
> > >
> > > ---
> > >
> > > ### 1. **Token Embedding Brings Polysemanticity?**
> > >
> > > To clarify, our claim is that adaptive token segmentation achieves **better** monosemanticity, rather than **perfect (100%)** monosemanticity. Here we provide additional analysis demonstrating that adaptive segmentation indeed **reduces feature variance**, which directly measures polysemanticity within each visual token.
> > >
> > > We extracted feature maps (in the shape of HxWxC) and generated token segmentation from 1k samples per dataset. Then we downsampled token segmentations (to HxW) to align it with the feature map and retrieved the corresponding token embeddings vectors (each with dimension of C). We calculate intra-feature variance within each segment to measure the polysemanticity. Results confirm that adaptive token segmentation consistently reduces feature variance compared to patch-based segmentation, with EPOC achieving the lowest variance:
> > >
> > > | Dataset (Embedding)  | Metric | Patch 10x10 (baseline) | Mask2Former (object-level) | EPOC (subobject-level) |
> > > |---|---|---|---|---|
> > > | ImageNet (VAE) | Variance | 0.504 |  0.473 | **0.442** |
> > > | ImageNet (DINOv2) | Variance | 6.287 |  6.104 | **6.019** |
> > > | | # Tokens |  100  |  5.0  |  50.5 |
> > > | Pixmo-cap (VAE)  | Variance | 0.560  |  0.474 | **0.465** |
> > > | Pixmo-cap (DINOv2) | Variance | 6.296 |  6.040  | **5.999** |
> > > | | # Tokens |  100  |  5.1 |  63.0  |
> > > | ShareGPT4V (VAE) | Variance | 0.579 |  0.465 | **0.457** |
> > > | ShareGPT4V (DINOv2) | Variance | 6.450  |  6.132 | **6.047** |
> > > | | # Tokens |  100  |  7.7 |  71.6 |
> > >
> > > Additionally, the method of **“deriving segmented token embedding in a bottom-up manner”** mentioned in your comment is already discussed in our submission. The limitations have been outlined in page 8, Section 6 of our submission:
> > >
> > > *“… **Feature-based methods, such as slot attention [1] and various token pooling methods [2-6], suffer from low-resolution feature maps which limit fine-grained segmentation, and unreliable segmentation quality in early training stages.** Our approach of using a separate segmentation model, EPOC, avoids these issues …*”
> > >
> > > > [1] Object-centric learning with slot attention. NeurIPS’20
> > > >
> > > > [2] Which tokens to use? investigating token reduction in vision transformers. ICCV’23
> > > >
> > > > [3] Vision transformers with mixed-resolution tokenization. ICCV’23
> > > >
> > > > [4] Token pooling in vision transformers. WACV’23
> > > >
> > > > [5] Efficient vision transformer via token merger. IEEE TIP, 2023
> > > >
> > > > [6] Vision transformer with super token sampling. CVPR’23
> > >
> > > ---
> > >
> > > ### 2. **Evaluating Token Segmentation via VQA?**
> > >
> > > The focus in Cambrian-1 study is to evaluate different **token embeddings** (e.g., CLIP, SigLIP, DINOv2, MAE) for their effectiveness in providing **general-purpose visual representations** across a wide range of **AI assistant tasks**, including knowledge, OCR, chart, etc.
> > >
> > > Our paper tackles a fundamentally different research question: whether **adaptive token segmentation** facilitates better learning of image understanding models. We clearly scoped our work and did not aim to present a state-of-the-art VLM. Instead, we provided thorough intrinsic and extrinsic evaluations to directly and empirically support our claim.
> > >
> > > The tasks of image captioning and VQA are closely related [7]. To illustrate, the following are some VQA examples from CLEVR, which shares the same visual source with our CLEVR-cap:
> > >
> > > **Examples in CLEVR (VQA)**
> > > - *Are there an equal number of large things and metal spheres? Yes.*
> > > - *What size is the cylinder that is left of the brown metal thing that is left of the big sphere? Big.*
> > > - *How many objects are either small cylinders or red things? 5.*
> > >
> > > **Example in CLEVR-cap**
> > > - *Total 10 objects: a small green rubber cylinder, a large blue metal cube, a small red rubber cylinder…*
> > >
> > > As it can be seen, both task formulations **assess overlapping capabilities**, including identification of attributes (shape, size, material, color), counting, and spatial relationships. Although VQA additionally evaluates logical reasoning capability required for producing the final answer, this reasoning capability is fundamentally **agnostic to image token segmentation**.
> > >
> > > We empirically demonstrate below that under the same LLM, the conclusions (i.e., performance ranking) drawn from token segmentation effectiveness using captioning datasets **translate directly and consistently to the VQA evaluation setting**:
> > >
> > > |  | CLEVR (VQA accuracy%) | CLEVR-cap (ratio of fully matched captions%) |
> > > |---|---|---|
> > > | **Patch** (10x10) | 21.5 - 3rd | 67.9 -3rd |
> > > | **Superpixel** (subobject-level) | 40.3 - 2nd | 75.5 - 2nd |
> > > | **EPOC** (subobject-level) | 48.0 - 1st | 80.9 -1st |
> > >
> > >
> > > > [7] All You May Need for VQA are Image Captions. NAACL’22
> > >
> > > ---
> > >
> > > Thank you again for your detailed feedback and constructive suggestions. We warmly welcome any further suggestions or questions you may have!

---

### Official Review · Reviewer_7brk · 2025-03-13

**Overall Recommendation:** 3

**Summary:**

Tokenization is an important step for any transformer-based model. For the vision transformers, it is often performed on a patch-level, where locally neighboring parts of the image are tokenized together in the form of small square patches. However, patch-based tokenization is not adaptive, that is it tokenizes any given patch equivalently. This work proposes an adaptive image tokenization strategy based on sub-object level features, aiming to address the drawbacks of the existing image tokenization strategies. The work further includes intrinsic (with respect to image) and extrinsic experimental analysis for highlighting the effectiveness of the proposed approach.

**Claims And Evidence:**

The proposed tokenization method claims to improve the patch-based tokenization with respect to token polysemanticity and token redundancy. In addition, it also claims to improve other adaptive tokenization methods with respect to computational efficiency and effective segmentation of subobject level regions.

With respect to the first claim, the authors present compelling verbal arguments, such as the issue of large patches encompassing multiple concepts (object classes, sub-object level details) and the redundancy coming from patchifying everywhere equivalently, e.g diving a background region into multiple tokens even though it is not very informative. In addition, the visualizations provided in Figure 3 and the quantitative analysis on Figure 4 demonstrate these issues more concretely. Given the fact that these arguments are also well-discussed and demonstrated in both the language and vision literatures, these claims are well-supported.

For the second claim, the authors present extrinsic evaluation results while utilizing an adjusted encoder-decoder model to accommodate for the dynamic nature of tokens following their tokenization process. While the discussion over the experiments is brief (and there is a lack of clarity in the presented figures, such as Figure 5), it can be seen that the proposed approach achieves better validation perplexity over certain alternatives while matching that of superpixel tokenization. In addition, the authors discuss the efficiency of the proposed approach and provide quantitative results backing this part of the claims. However, as detailed in the methods and evaluation criteria, there are several points that are preventing me from stating that this claim is overall well-supported.

**Essential References Not Discussed:**

N/A

**Experimental Designs Or Analyses:**

The experimental analysis presented seems sound, though most of the results follow a rather ad-hoc structure for the work. Specifically, the instrinsic evaluation only discusses the edge-related metrics while omitting mask-level ones. While this is partially understandable given the nature of the work, it is still expected that the patch-based tokenization methods or those which do not necessarily rely on edges may not perform as the presented method. As detailed under the methods and evaluation criteria part, extrinsic evaluation also contains several ad-hoc choices.

**Methods And Evaluation Criteria:**

- While it is understandable that it is non-trivial to utilize dynamically tokenized patches with off-the-shelf VLMs, I find the proposed methodology in Section 5 to be confusing, especially the rationale behind using a completely separate visual encoder and a VLM on top. Given the plethora of models with their own visual encoder and language decoder on top [A, B, C, D], I am not fully sure why the authors performed the analysis the way it is, instead of performing small adjustments on the aforementioned models.

- In addition, only the validation perplexity is presented for the utilized benchmarks. However, various other metrics associated with these benchmarks could have been reported (e.g BLEU scores [E], CIDEr [F] for captioning-based ones, classification accuracy for Imagenet) but were instead omitted.

Other than these points, the methodology seems reasonable.

[A] Liu, Haotian, et al. "Visual instruction tuning." Advances in neural information processing systems 36 (2023): 34892-34916.

[B] Li, Junnan, et al. "Blip: Bootstrapping language-image pre-training for unified vision-language understanding and generation." International conference on machine learning. PMLR, 2022.

[C] Yu, Jiahui, et al. "Coca: Contrastive captioners are image-text foundation models." arXiv preprint arXiv:2205.01917 (2022).

[D] Tschannen, Michael, et al. "Image captioners are scalable vision learners too." Advances in Neural Information Processing Systems 36 (2023): 46830-46855.

[E] Papineni, Kishore, et al. "Bleu: a method for automatic evaluation of machine translation." Proceedings of the 40th annual meeting of the Association for Computational Linguistics. 2002.

[F] Vedantam, Ramakrishna, C. Lawrence Zitnick, and Devi Parikh. "Cider: Consensus-based image description evaluation." Proceedings of the IEEE conference on computer vision and pattern recognition. 2015.

**Other Comments Or Suggestions:**

- Minor typo on L105: toknizers -> tokenizers
- Minor typo on L369: coverts -> converts
- The Figures 4 and 5 are not very easy on the eye and do not quickly convey the message of the work. I think that the work could benefit from moving some of the sub-figures to Appendix and emphasizing the most significant results from them.

**Other Strengths And Weaknesses:**

The proposed idea based on the watershed algorithm and edge detection is interesting and novel for image tokenization. The goals of the work are also significant and are very timely. With respect to clarity, I think that the work could benefit from a diagram or a more fluent description of the extrinsic evaluation set-up, otherwise the thoughts are presented fluently.

**Questions For Authors:**

-  I am not sure what the Figure 6 is presenting: Is it just the convergence with respect to training loss? Would not it benefit from the presentation of generalization performance too?
- As a very minor question, where do you think this work stands in comparison to [A] and [B]?


[A] Kim, Young Kyung, J. Matías Di Martino, and Guillermo Sapiro. "Vision transformers with natural language semantics." arXiv preprint arXiv:2402.17863 (2024).

[B] Aasan, Marius, et al. "A Spitting Image: Modular Superpixel Tokenization in Vision Transformers." arXiv preprint arXiv:2408.07680 (2024).

**Relation To Broader Scientific Literature:**

The work aims to tackle a timely problem in the vision literature, namely adaptive tokenization for vision transformers, for allowing efficient allocation of computational resources to more informative areas of images. Given the prominence of recent similar approaches in the language domain [A], if evidenced strongly, the work could be interesting to the broader transformers/deep learning community.


[A] Pagnoni, Artidoro, et al. "Byte latent transformer: Patches scale better than tokens." arXiv preprint arXiv:2412.09871 (2024).

**Theoretical Claims:**

The paper does not include detailed theoretical discussions or theorems. However, the intuitions and motivations behind the work are clearly explained.

---

> ### Author Rebuttal · Authors · 2025-04-01
>
> Thank you very much for your detailed and comprehensive review. We sincerely appreciate the considerable effort and depth of analysis you provided! Below, let us address each of your concerns point-by-point and outline concrete steps we will take to improve the final manuscript:
>
> ---
>
> ### 1. **VLM Architecture in Extrinsic Evaluation** (in “Methods And Evaluation Criteria”)
>
> > **TL;DR**  Our approach is precisely a **minimal adjustment** of the LLaVA architecture. The modification involves adding positional embeddings to **support adaptive token segmentation**.
>
> We appreciate your comment and realize that our original description might have caused confusion. To clarify, our VLM architecture directly builds upon the well-established **LLaVA framework** (visual encoder → MLP connector → LLM). We do not introduce any new paradigm, such as *"separate visual encoder and a VLM on top"* as suggested. In the final version, we will explicitly emphasize this fact in Sec 5.
>
> The sole reason for the modification is to handle **adaptive token segmentation**, which inherently involves variations in token size, position, and arrangement—an aspect not supported by existing architectures. Apart from this, our training settings and datasets (e.g., using ShareGPT-4V, Pixmo-cap) **align well with common practices** in VLM pretraining.
>
> ---
>
> ### 2. **Perplexity-based Metric in Extrinsic Evaluation** (in “Methods And Evaluation Criteria”)
>
> We appreciate this important suggestion. To clarify, we used validation perplexity mainly to **maintain consistency and clarity** across different datasets. We fully agree that additional metrics are valuable. For all data points in Fig. 5, we calculated their accuracy or BLEU as suggested, and measured their correlation with perplexity. As below, two metrics **strongly correlate** with each other. We will include these results in the appendix of the final manuscript:
>
> | |CLEVR-cap | ImageNet | ShareGPT4v | Pixmo-cap |
> |---|---|---|---|---|
> | Metric  | Accuracy  | Accuracy  | BLEU  | BLEU |
> | Pearson Correlation with Perplexity | -0.86$^*$  | -0.97  | -0.93  | -0.91 |
> | p-value | p < 0.0001 | p < 0.0001 | p < 0.0001 | p < 0.0001 |
>
> $^*$The correlation on CLEVR-cap seems weaker compared to others. This is because they exhibit a log-linear relationship according to our visualization. The correlation between perplexity and log(accuracy) is -0.95.
>
> ---
>
> ### 3. **Boundary-based Metric in Intrinsic Evaluation**  (in “Experimental Designs Or Analyses”)
>
> > **TL;DR** Mask-based metrics are fundamentally unsuitable for evaluating image tokenizers. Boundary-based metrics accurately assess tokenization quality in **class-agnostic** and **multi-granularity** settings and have been widely adopted in CV and NLP.
>
> Mask-based metrics, such as mIoU, depend heavily on clearly defined semantic classes, which makes them inappropriate for evaluating our tokenizer. They also struggle when segmentation granularities differ between predictions and ground truth. Accurately subdividing an annotated object into several meaningful subparts may produce **misleadingly low IoU scores**.
>
> Boundary-based metrics naturally overcome these limitations. These metrics have also long been standard in boundary detection literature and have been used by SAM-related studies. They are also applied in NLP tokenizer evaluation, assessing the alignment to linguistic morphological boundaries.
>
> Additionally, the **Token Monosemanticity Score** is essentially a mask-based metric (see response 1 to reviewer Ev7G).
>
> ---
>
> ### 4. **Reporting Training Loss in Fig.6.** (in “Questions For Authors”)
>
> The current Fig.6 is presenting **average training loss**, which is an indicator of convergence speed. To align with Fig.5, we will update it to report **validation perplexity** following your suggestion. In fact, the two metrics closely correlate with each other, as it can be seen in Fig. 15. With validation performance in Fig.6, the relative performance relationship stays **exactly the same** and the conclusions in Sec. 5.3 remains unchanged.
>
> ---
>
> ### 5. **Relation to Recent Works**  (in “Questions For Authors”)
>
> Thank you for highlighting these strongly related recent works. The submission already included [B] (Aasan et al., 2024), and we will cite and discuss [A] in our final version. Both studies explore **adaptive token segmentation**, closely aligning with our research, yet with important differences:
>
> - Compared to EPOC, the superpixel approach in [B] performs bottom-up pixel grouping without **semantic understanding**, and SAM-based segmentation in [A] cannot ensure **efficient** and **panoptic** segmentation. They are already involved as baseline methods in our paper.
>
> - These works focus on **ViTs** and **image classification**, while our approach extends to **VLMs** and evaluates performance on the more challenging task of **detailed image captioning**.
>
> ---
>
> Once again, thank you very much for your comprehensive and insightful comments!

---

> > ### Comment · Reviewer_7brk · 2025-04-07
> >
> > I apologize to the authors for my late reply and appreciate their detailed response. In particular, I appreciate that the authors presented brief results regarding how BLEU correlates with their measures and further clarifications on the architecture and relations to recent works. I encourage the authors to integrate these to their work as well.
> >
> > While I still have concerns (e.g correlation is good, but insufficient, what about the actual numbers for BLEU4/CIDEr?), the rebuttal presented by the authors both to my review and other reviewers' is sufficient to tilt my decision towards acceptance. Accordingly, I will be raising my score.

---

> > > ### Author Response · Authors · 2025-04-07
> > >
> > > Thank you very much for your thoughtful follow-up and for reconsidering our manuscript! We greatly appreciate your valuable suggestions. As recommended, we will integrate detailed results (including many exact numbers, which cannot be fully enumerated in the response above due to the 5k character limit) into the final manuscript to support our findings.
> > >
> > > Your feedback has significantly helped us improve the clarity and rigor of our paper. We're delighted that our rebuttal addressed your concerns and greatly appreciate your updated evaluation. If you have any further comments or suggestions, please feel free to let us know—we warmly welcome additional discussion!

---

### Official Review · Reviewer_avnM · 2025-03-13

**Overall Recommendation:** 4

**Summary:**

This paper proposes sub object-level image tokenization, which tokenize image based on the morphological structure of the image. Compared to other potential subobject tokenizers, EPOC improves efficiency. Experiments on multiple VLMs demonstrate the advantages of the subobject tokenizer.

## update after rebuttal
During the rebuttal, the authors provided additional experiments that addressed my and other reviewers' concerns. I keep my score as accept.

**Claims And Evidence:**

All the claims made in the submission supported by clear evidence.

**Essential References Not Discussed:**

N/A

**Experimental Designs Or Analyses:**

Please see Methods And Evaluation Criteria section

**Methods And Evaluation Criteria:**

Regarding the question on extrinsic evaluation, why is the comparison only made on caption data and not on general VQA benchmarks? Is it because it doesn't perform as well?

The second question concerns Section 5.3, "Results and Discussions - Compatibility to Different Token Embeddings." It mentions that when using the subobject tokenizer, the DINO model performs better than CLIP. However, in mainstream models, DINO does not outperform CLIP. The explanation provided in the paper is that this is due to the lower resolution. Does the author believe that, for now, the subobject tokenizer still has limited practical value?

**Other Comments Or Suggestions:**

N/A

**Other Strengths And Weaknesses:**

The writing is clear and convincing.

**Questions For Authors:**

In my view, under image-LLM, the image tokenizer doesn't really need significant improvements, as even ultra-high resolutions are still manageable for current GPUs. I believe the subobject tokenizer is more suitable for use in video.

**Relation To Broader Scientific Literature:**

N/A

**Theoretical Claims:**

yes, I check the correctness of proof in the paper.

---

> ### Author Rebuttal · Authors · 2025-04-01
>
> Thank you for your encouraging review and thoughtful questions. Your recognition of the strengths of our method is greatly appreciated. Below are our clarifications on your valuable questions:
>
> ---
>
> ### 1.**Extrinsic evaluations on caption data rather than general VQA** (in “Methods And Evaluation Criteria”)
>
> Our main goal was to isolate image understanding performance clearly from complex reasoning tasks. Standard VQA datasets tend to fuse perception with significant **knowledge** and **reasoning** capabilities of LLMs. For example, questions like “Where is he looking?”, or “What time is it on the clock?” which commonly occurs on VQA data requires the VLM to understand concepts like “third-person pronoun”, “sense of direction”, and “time” which can potentially **confound the assessment of the effectiveness of adaptive token segmentation**. Thus, we focused on detailed captioning datasets to precisely evaluate token-level perception capability.
>
> ---
>
> ### 2. **Practical value regarding the comparison between DINO and CLIP** (in “Methods And Evaluation Criteria”)
>
> The observed advantage of DINO over CLIP in our experiments primarily results from the higher resolution and dense supervision available in DINO embeddings. CLIP, especially the original OpenAI version, lacks resolution flexibility with a 7x7 feature map, limiting fine-grained perception that is essential for subobject tokenization. We agree CLIP still holds significant practical value, especially when high-resolution features are integrated, which can further enhance subobject tokenization.
>
> ---
>
> ### 3. **Future direction towards video tokenization** (in “Questions For Authors”)
>
>
> Indeed, as we stated in Appendix E, subobject-level tokenization's greatest potential may lie in video understanding, where computational efficiency and token management are even more critical due to the temporal dimension. Our work provides foundational insights that naturally extend to video data, and we consider this a key future direction.
>
> Thank you once again for your valuable input and for clearly identifying the broader applications of our method.

---

> > ### Comment · Reviewer_avnM · 2025-04-02
> >
> > I have read the other reviewers' comments (including the negative remarks by Reviewer 7brk D5bb) as well as the authors' rebuttal. While I acknowledge some of the shortcomings raised, I still find the positive aspects outweigh the negatives. In particular, I appreciate the proposed sub-object-level image tokenization approach. I thank the authors for their response and maintain my acceptance rating.

---

> > > ### Author Response · Authors · 2025-04-03
> > >
> > > Thank you very much for your thoughtful consideration and continued support for our paper! We are trying our best to provide additional evidence to address the concerns raised by other reviewers. Your positive feedback has greatly encouraged us, and we warmly welcome any further suggestions or discussions!

---

### Official Review · Reviewer_Ev7G · 2025-03-13

**Overall Recommendation:** 4

**Summary:**

This paper introduces **Subobject-level Image Tokenization**, a novel adaptive image tokenization strategy inspired by subword tokenization in NLP. Previous patch-based image tokenization methods suffer from inefficiencies and polysemanticity. To address these limitations, the paper proposes a new tokenizer called **Efficient and PanOptiC (EPOC)**, which combines boundary detection and watershed segmentation to guarantee comprehensive segmentation and computational efficiency.

Main Contributions:
1. **Panoptic Segmentation Improvement** The proposed  EPOC integrates boundary detection and watershed segmentation.
2. **Computational Efficiency**  The proposed  EPOC achieves better token efficiency.

Main Results:
1. Intrinsic Evaluation: Evaluations on five datasets demonstrate that EPOC tokens align well with human semantic annotations, achieving higher monosemanticity scores.
2. Extrinsic Evaluation(VLMs): EPOC-based tokenization achieves faster convergence, improved generalization, and better token efficiency.

**Claims And Evidence:**

The claims made by the paper are strongly supported by a clear proposed method (EPOC), well-designed experiments (intrinsic and extrinsic evaluations), and quantitative and qualitative experimental results.

**Essential References Not Discussed:**

The essential references are well discussed in Section 6.

**Experimental Designs Or Analyses:**

The experiments in this paper are well-designed.
1. Intrinsic evaluations are performed across five datasets. The authors focus on boundary precision-recall metrics and monosemanticity score.
2. Extrinsic evaluations are performed on four datasets which are high-quality and widely-used. Besides, the authors use different feature embeddings to demonstrate the robustness of the proposed method.

**Methods And Evaluation Criteria:**

The proposed methods make clear sense for the problem of image tokenization. The paper carefully compares the proposed methods with the previous one (patch-based tokenization and object-level segmentation).

**Other Comments Or Suggestions:**

Here are some comments and suggestions for the authors:

- Highly recommend reorganizing the highlighted or underlined words in section 5.2.
- As for Figure 5, I highly recommend consistently plotting with grids.

**Other Strengths And Weaknesses:**

Paper Strengths:
1. The paper is well-written and easy to follow.
2. The paper introduces a novel analogy from NLP to CV, which is conceptually innovative.
3. The proposed EPOC achieves computational efficiency, which could lead to high-resolution images with fewer computational resources.

Weaknesses:
1. Limited novelty.
2. Limited ablation studies on position and content embeddings.

**Questions For Authors:**

Questions for authors:

1. Can the authors formulate the token monosemanticity score?
2. Why do you choose **SegFormer** as a boundary prediction model? Did the authors try to use other models to perform the same role, and what could be the performance? Can the authors provide some experiment with other boundary prediction models?
3. Can the authors provide some ablation on position and content embeddings?
4. Can the EPOC be scaled up?

**Relation To Broader Scientific Literature:**

1. This paper is related to vision transformers using patch-based image tokenization.
2. This paper is related to the image segmentation models.

**Theoretical Claims:**

There are no theoretical claims that require correctness in this paper.

---

> ### Author Rebuttal · Authors · 2025-04-01
>
> Thank you for your thorough and insightful review. Below, we provide detailed responses addressing your specific questions and suggestions:
>
> ---
>
> ### 1. **Formulating Token Monosemanticity Score** (Question 1)
> Yes. Below is the explicit formal definition, which will be added to the final version:
>
> **Definition 1 (Token Monosemanticity Score):** Given an image \\( \\mathbf{X}\\in\\mathbb{R}^{H\\times W\\times 3} \\), a predicted token segmentation \\( \\mathbf{M}\\in\\{0,\\dots,N-1\\}^{H\\times W} \\), and a ground-truth segmentation \\( \\mathbf{M}^{\\ast}\\in\\{0,\\dots,K-1\\}^{H\\times W} \\), define the \\( i \\)-th predicted token \\( \\mathbf{t}_i \\) as the set of pixel coordinates assigned token index \\( i \\), i.e., \\( \\mathbf{t}_i=\\{(h,w)\\mid \\mathbf{M}(h,w)=i\\} \\).
>
> We say a token \\( \\mathbf{t}_i \\) is **monosemantic** if it lies entirely within exactly one ground-truth semantic region. Formally, the indicator function of token monosemanticity is defined as follows:
>
> \\[
> \\mathbb{I}_{\\text{mono}}(\\mathbf{t}_i)=
> \\begin{cases}
> 1, & \\text{if } \\exists k \\text{ such that } \\forall (h,w)\\in \\mathbf{t}_i, \\mathbf{M}^{\\ast}(h,w)=k \\\\
> 0, & \\text{otherwise.}
> \\end{cases}
> \\]
>
> Then, the **Token Monosemanticity Score** is defined as the fraction of all predicted tokens that are monosemantic:
>
> \\[
> \\text{monosemanticity}(\\mathbf{M}, \\mathbf{M}^{\\ast})=\\frac{1}{N}\\sum_{i=0}^{N-1}\\mathbb{I}_{\\text{mono}}(\\mathbf{t}_i).
> \\]
>
> Intuitively, this metric quantifies how effectively the predicted segmentation avoids polysemantic tokens, which span multiple semantic regions.
>
> ---
>
> ### 2. **Boundary Detector Backbone** (Question 2 & 4)
>
> We chose SegFormer primarily due to its popularity, simplicity, and computational efficiency. However, EPOC is definitely not confined to SegFormer. Boundary detection itself is a historically well-studied and relatively straightforward task, effectively handled by various convolutional or Transformer backbones [1-4].
>
> > [1] DeepContour: A deep convolutional feature learned by positive-sharing loss for contour detection. CVPR’15
> >
> > [2] Richer convolutional features for edge detection. CVPR’17
> >
> > [3] EDTER: Edge detection with transformer. CVPR’22
> >
> > [4] DiffusionEdge: Diffusion Probabilistic Model for Crisp Edge Detection. AAAI’24
>
> To directly address your questions regarding scalability and alternative backbones, we conducted an additional scaling experiment comparing the original SegFormer-b0 to a substantially larger SegFormer-b5. Both models were trained on SA-1B dataset for 2 epochs, and the followings are performance evaluation (boundary recall) results:
>
> | Model | Parameters  | Object-level (ADE20k) | Subobject-level (PPP) |
> |---|---|---|---|
> | SegFormer-b0 (Submission) | 3.7M  | 58.88 | 69.22 |
> | SegFormer-b5 (New)  | 87.5M (x24) | 65.73 (+6.85%)  | 71.46 (+2.24%)  |
>
> These results clearly show that while EPOC can indeed scale up with larger models, the performance gains are marginal relative to the substantial increase in computational overhead. This further justifies our original choice of the efficient SegFormer-b0.
>
> ---
>
> ### 3. **Ablation on Position and Content Embeddings** (Question 3)
>
> We agree that an ablation study on position and content embeddings provides valuable insights into how VLMs utilizing adaptive token segmentation interpret images. We conducted an experiment with our EPOC-based VLM trained on CLEVR-cap, where we shuffled positional or content embeddings across visual tokens, and then evaluated the impact on performance.
>
> Since CLEVR-cap is synthetic and the captions follow a structured template—e.g., *“Total 5 objects: a small gray metal cube, a small red rubber sphere, …”*—it allows us to parse the generation and calculate average accuracy (%) for individual attributes. The results are summarized below:
>
> |  | Count | Size | Color | Material | Shape |
> |---|---|---|---|---|---|
> | No Ablation | 57.0 | 71.4 | 59.3 | 77.7 | 69.3  |
> | Shuffle Position Embeddings | 35.0 | 49.9 | 16.8 | 47.3  | 50.5  |
> | Shuffle Content Embeddings | 48.0 | 66.4 | 20.8  | 52.0  | 59.3  |
> | Chance | 10.0 | 50.0 | 12.5 | 50.0 | 33.3 |
>
> These results show that both positional and content embeddings contribute to accurately capturing semantic attributes and correctly associating them with their corresponding objects, confirming their complementary roles in adaptive token segmentation-based image understanding.
>
> ---
>
> ### 4. **Comments and Suggestions on Section 5.2 and Fig. 5**
>
> Thank you very much for these practical suggestions. Indeed, Section 5.2 is densely packed due to the strict page limits of the ICML submission format. The final version is allowed an extra page, enabling us to reorganize the presentation. Additionally, we will ensure Figure 5 includes consistent grid lines across all subplots.
>
> ---
>
> Finally, thank you again for your constructive and detailed feedback, which significantly improves the quality and clarity of our paper.

---

> > ### Comment · Reviewer_Ev7G · 2025-04-07
> >
> > I apologize to the authors for my late reply and appreciate their detailed response. I appreciate that the authors respond to my main concerns about the paper., especially the ablation on position and content embeddings.  I will be raising my score.

---

> > > ### Author Response · Authors · 2025-04-07
> > >
> > > Thank you very much for your thoughtful follow-up and for reconsidering our manuscript! We deeply appreciate your acknowledgment of our efforts! If you have any additional suggestions or thoughts, please feel free to let us know—we warmly welcome further discussion!

---

### Decision · Program_Chairs · 2025-05-01

**Decision:**

Accept (poster)

**Comment:**

This paper introduces a novel adaptive image tokenization strategy inspired by subword tokenization in NLP to align with visual morphology, improve token quality and efficiency, and enable vision-language models to converge faster and generalize better using fewer tokens.

It has received 4 reviews, with 2x accepts, 1x weak accept, and 1x weak reject.  Authors have provided detailed rebuttals which are successful at raising reviewers' final ratings.  Reviewers find the paper well written, conceptually innovative, and delivered practical efficiency.

The negative reviewer had a lingering concern on the performance on VQA, which authors did provide additional results in a further response (after the reviewer's rebuttal acknowledgement).

There is also an interesting discussion on the separation of token segmentation and token embedding that the authors should include in the final version.   The following work is highly relevant -- there are several other recent related works in computer vision that deal with end-to-end adaptive segment tokens such as HGFormer, SSN, SimSeg etc:

Learning Hierarchical Image Segmentation For Recognition and By Recognition, Tsung-Wei Ke, Sangwoo Mo, Stella X. Yu, ICLR 2024
(Earlier version: CAST: Concurrent Recognition and Segmentation with Adaptive Segment Tokens, ArXiv 2210.00314, 2022)

It would be helpful to add results on the more challenging Cityscapes dataset.